# A dorsomedial prefrontal cortex-based dynamic functional connectivity model of rumination

Jungwoo Kim [1,2,3,12], Jessica R. Andrews-Hanna[4,5,12], Hedwig Eisenbarth [6], Byeol Kim Lux [1,2,7], Hong Ji Kim[1,2,3], Eunjin Lee [1,2,3], Martin A. Lindquist[8], Elizabeth A. Reynolds Losin[9,10], Tor D. Wager [7,13] & Choong-Wan Woo [1,2,3,11,13]

Rumination is a cognitive style characterized by repetitive thoughts about one's negative internal states and is a common symptom of depression. Previous studies have linked trait rumination to alterations in the default mode network, but predictive brain markers of rumination are lacking. Here, we adopt a predictive modeling approach to develop a neuroimaging marker of rumination based on the variance of dynamic resting-state functional connectivity and test it across 5 diverse subclinical and clinical samples (total $n = 288$). A whole-brain marker based on dynamic connectivity with the dorsomedial prefrontal cortex (dmPFC) emerges as generalizable across the subclinical datasets. A refined marker consisting of the most important features from a virtual lesion analysis further predicts depression scores of adults with major depressive disorder ($n = 35$). This study highlights the role of the dmPFC in trait rumination and provides a dynamic functional connectivity marker for rumination.

Individuals who ruminate are plagued by persistent negative, self-reflective thoughts. Such thoughts are focused on the causes, consequences, and symptoms of distress[1], ostensibly to help people make sense of the situations that are causing distress and formulate action plans. However, when productive actions are not available, people can get stuck in a cycle of rumination that can aggravate and prolong distress[2] and thus contribute to multiple psychopathologies such as depression and anxiety[3]. Importantly, rumination also often emerges as an early risk factor for depression, highlighting the need for improved subclinical detection and intervention before the onset of

clinical episodes[4]. Resting-state fMRI (rsfMRI) holds great promise as a tool for developing objective brain-based markers of rumination, particularly as rest is a natural condition that may facilitate ruminative thinking. Although prior studies have pointed to altered patterns of rsfMRI connectivity in individuals who ruminate, precisely specified models that can be applied to new individuals to capture and predict rumination remain lacking. In other areas, whole-brain predictive connectivity-based models that take dynamic features of rsfMRI into account (e.g., temporal fluctuations in functional connectivity) have produced generalizable neuromarkers for depression, pain, attention,

[1]Center for Neuroscience Imaging Research, Institute for Basic Science, Suwon, South Korea. [2]Department of Biomedical Engineering, Sungkyunkwan University, Suwon, South Korea. [3]Department of Intelligent Precision Healthcare Convergence, Sungkyunkwan University, Suwon, South Korea. [4]Department of Psychology, University of Arizona, Tucson, AZ, USA. [5]Cognitive Science, University of Arizona, Tucson, AZ, USA. [6]School of Psychology, Victoria University of Wellington, Wellington, New Zealand. [7]Department of Psychological and Brain Sciences, Dartmouth College, Hanover, NH, USA. [8]Department of Biostatistics, Johns Hopkins University, Baltimore, MD, USA. [9]Department of Psychology, University of Miami, Miami, FL, USA. [10]Department of Biobehavioral Health, Penn State University, State College, PA, USA. [11]Life-inspired Neural Network for Prediction and Optimization Research Group, Suwon, South Korea. [12]These authors contributed equally: Jungwoo Kim, Jessica R. Andrews-Hanna. [13]These authors jointly supervised this work: Tor D. Wager, Choong-Wan Woo. ✉ e-mail: tor.d.wager@dartmouth.edu; waniwoo@g.skku.edu

and other outcomes[5–8]. Our goal here was to extend this approach to rumination.

Among large-scale resting-state networks previously linked to rumination, the brain's default mode network (DMN) has been reported most consistently[9]. Theories of DMN function highlight its role in internally oriented processes thought to be dysfunctional in ruminative individuals[10], spanning self-referential thought[11], autobiographical memory[12], emotional experience[13], and more. The DMN has also recently been suggested as a hub for constructing representations of the self in relation to situational contexts, integrating multimodal information from one's body state (via interoceptive and autonomic pathways), memory systems, and predictions about future states into an internal conceptual model[14,15]. Such hypothesized functions may also explain the involvement of the DMN in depression and numerous other mental health disorders linked to dysfunctional thoughts and emotions[16–18].

Critically, the DMN is thought to be heterogeneous in anatomy and function[19,20], with some hypotheses suggesting interacting subsystems that support the processes by which thoughts and emotions are construed[21]. For example, a medial temporal system may support mental experiences at low levels of construal—a so-called "mind's eye," imbuing mental simulations with rich spatial, temporal, and perceptual details. In contrast, a dorsal medial system may support a "mind's mind" form of cognition, whereby high-level, reflective processes allow us to consider the broader implications and significance of our thoughts, emotions, and external stimuli[21]. Interpreted in this framework, a dorsal medial system is the most reasonable candidate important for the repetitive high-level appraisal that characterizes ruminative thinking. Particularly, the dorsal medial prefrontal cortex (dmPFC) has been suggested to be a core brain region for rumination. It has been considered a "dorsal nexus" that serves as a core that modulates the connectivity related to depression[22], and the heightened connectivity of the dmPFC was a unique feature of major depressive disorder (MDD) compared to other mental disorders[23] (see also refs. [24,25]). We, therefore, hypothesized that dmPFC connectivity may be important for trait rumination. However, there have also been some inconsistent reports in the literature. For example, the static functional connectivity strength within the dorsal medial system was increased[16] or decreased[26] in individuals with MDD (see also refs. [27,28]). Collectively, these findings highlight the need for adopting a predictive modeling approach, which could provide more reliable results[29]. Furthermore, in addition to the dmPFC, we also tested other brain regions across the DMN to minimize the potential bias in our findings.

Central to the ruminative style are altered cognitive and affective dynamics, whereby negative self-focused thoughts are difficult to let go of and consequently persist over time[30]. In light of this important feature of rumination, we hypothesized that the variance of dynamic functional connectivity would serve as an important predictor of rumination. Static or averaged dynamic DMN connectivities would reflect whether the connections between regions are high or low during the resting-state scan and have been one of the key functional brain features for characterizing multiple clinical conditions[16,22,23]. However, such static connectivity measures cannot capture how stable or variable they are over time, which we hypothesized to be a key feature of rumination. Despite the importance of the connectivity variance, only a few studies have explored the relevance of such measures to rumination[31,32]. In addition, no studies of which we are aware have developed dynamic connectivity-based predictive models that can predict trait rumination in new individuals. Such predictive models could provide a direct window into depression-relevant brain processes without the filtering inherent in self report. Additionally, an identification of the key neural mechanism comprising a seed-based predictive model may not only serve as a potential tool for identifying individuals at risk for mental illness, but it may also offer guidance on how pharmacological or psychological interventions can be applied to modulate targeted brain regions[5,33].

In this work, we apply dynamic connectivity-based predictive modeling to five independent datasets (including four subclinical datasets and one publicly available clinical dataset) to answer the following questions: (1) Can we develop a predictive model of rumination generalizable across multiple studies using rsfMRI-based dynamic functional connectivity within subclinical samples? (2) Which functional connections are important contributors to the prediction of rumination? and 3) Can the model predict the depression scores of a clinical sample? To answer these questions, we use three independent rsfMRI datasets from subclinical samples for model training, validation, and independent testing ($n = 84$, 61, and 48, respectively; Fig. 1a). Within the Study 1 dataset (i.e., training data), we develop multiple models predictive of the three subscales of the Ruminative Response Scale (RRS)[34]—i.e., brooding, depressive rumination, and reflective pondering. We use 20 predefined DMN seeds[35] to calculate seed-based dynamic conditional correlations (DCC)[36], which allow us to assess dynamic connectivity between each seed and 280 brain regions. We use the variance of the DCC values as input features to predict RRS subscale scores, resulting in a total of 60 models (20 seeds × 3 subscales). We then test the models on Study 2 and 3 datasets (i.e., validation and independent test datasets) to identify predictive models that generalize across multiple datasets. From the same datasets, we use the virtual lesion method to identify the important features from the original full model. Finally, we test the refined model comprised of the identified important features on a separate clinical dataset (Study 4) consisting of 35 adults with MDD to predict their depression scores measured with the BDI-II.

Overall, we identify the dynamic connectivity-based predictive model of rumination involving the dmPFC that holds the potential for evaluating rumination in subclinical and clinical populations. This study also expands our understanding of the role of the dmPFC and functionally connected regions in psychological processes central to trait rumination.

## Results
### Model development and independent testing
As shown in Fig. 1a, we first calculated seed-based DCC values using 20 DMN subregions[35] as seeds. We used the variance of seed-based DCC values as input features (a total of 280 features) to separately predict three subscales of RRS (i.e., brooding, depressive rumination, and reflective pondering) with Lasso regression, resulting in 60 whole-brain predictive models based on the variance of dynamic connectivity (20 seeds × 3 subscales). Among these 60 predictive models, we chose models that showed significant prediction performance from leave-one-participant-out cross-validation and tested them on an independent validation dataset (Study 2, $n = 61$). Seven predictive models showed significant prediction performance at this stage with $q < 0.05$, false discovery rate (FDR) corrected ($p < 0.006$). The selected models included the predictive models for the brooding subscale based on the ventromedial prefrontal cortex (vmPFC), left posterior cingulate cortex (PCC), and left temporal pole (TempP) seed regions, for the depressive rumination subscale based on the dmPFC seed region, and for the reflective pondering subscale based on the right parahippocampal cortex (PHC), right temporoparietal junction (TPJ), and right TempP seed regions. We then tested these models on an independent test dataset (Study 3, $n = 48$) and selected the predictive model that showed generalizable prediction performance. At this stage of independent testing, only the dmPFC-based predictive model of the depressive rumination subscale showed a significant generalization. The dmPFC-based model showed cross-validated prediction performance of $r = 0.342$ ($p = 0.001$, one-sided permutation test, 95% confidence interval (CI) [0.139, 0.574]) in the training dataset, $r = 0.240$ ($p = 0.037$, one-sided permutation test, 95% CI [−0.013, 0.502]) in the

validation dataset, and $r = 0.288$ ($p = 0.025$, one-sided permutation test, 95% CI [0.004, 0.589]) in the independent test dataset (Fig. 1b). When we removed one outlier in the independent test dataset, the dmPFC-based model still showed significant prediction performance of $r = 0.276$ ($p = 0.028$, one-sided permutation test, 95% CI [−0.012, 0.579]). The overall prediction results are shown in Table 1 and Supplementary Table 2.

We also trained and tested the models with static connectivity as input features, but none of the models survived (Supplementary Table 3), suggesting that dynamic functional connectivity is more sensitive to individual differences in rumination than static connectivity. In addition, to examine the robustness of our results, we repeated our analysis by shuffling and combining the training, validation, and testing datasets. When we trained predictive models with the Study 2 or Study 3 dataset alone, we could not replicate the results (Supplementary Tables 4, 5). However, when we trained the models with the combined dataset of Studies 2 and 3 ($n = 109$) and tested the model on the Study 1 dataset, we were able to replicate the original results—the dmPFC-based predictive model showed significant prediction performance in both training and testing datasets, but it did so only when the number of predictors was greater than 80 (Supplementary Fig. 4). These additional analyses suggest that the successful prediction of rumination requires at least a certain number of predictors. In addition, to further investigate the impact of the number of features, we compared the results of using the number of features same as the original model (i.e., $n_{\text{feature}} = 84$; Supplementary Table 6)

with the results of using the maximum possible number of features (i.e., $n_{\text{feature}} = 109$; Supplementary Table 7). In both cases, only the dmPFC-based predictive model of depressive rumination showed significant predictions across training and testing datasets.

## Characterizing the dmPFC-based predictive model of depressive rumination

To better understand the model, we examined the model weights (Fig. 2a). The dmPFC-based predictive model of depressive rumination included 84 non-zero predictive connection weights, which consisted of 38 positive and 46 negative weights. Regions with positive predictive weights indicate that more variable functional connectivity (i.e., higher temporal variance) between the dmPFC and the regions is predictive of higher depressive rumination scores, whereas negative weights indicate that less variable (or more stable) functional connectivity (i.e., lower temporal variance) with the dmPFC is predictive of higher depressive rumination scores. Among 38 regions with positive weights, 8 regions were subcortical regions, including subregions of basal ganglia, hippocampus, and thalamus. Six regions fell within the dorsal attention network, including subregions of the right inferior temporal gyrus, left inferior parietal lobule, and bilateral superior parietal lobules. Six regions fell within the frontoparietal network, including subregions of the right medial frontal gyrus, right orbital gyrus, and right insula. And, finally, five regions fell within the visual network, including subregions of the bilateral lateral occipital complex and the left medioventral occipital

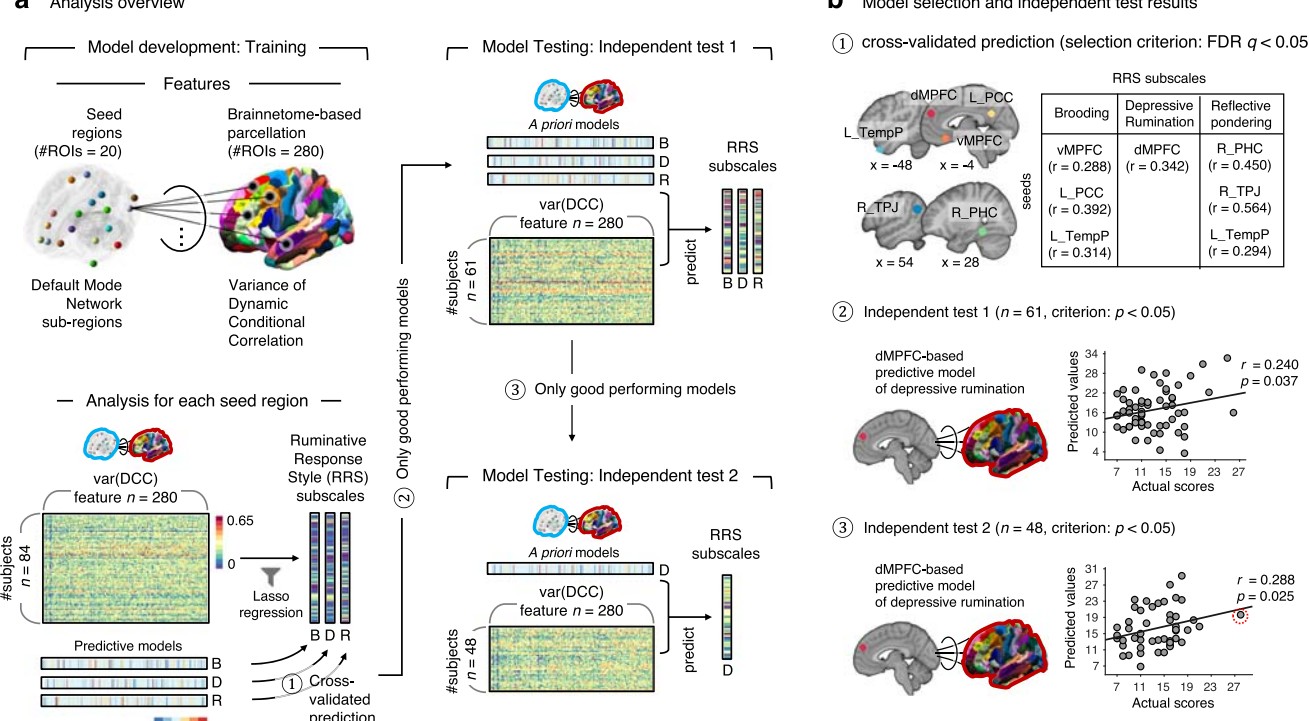

**Fig. 1 | Analysis overview and prediction results. a** For the model development, we first predefined 20 seed regions within the DMN based on ref. 35. We then calculated the Dynamic Conditional Correlation (DCC) between each seed region and 280 Brainnetome-based parcels using rsfMRI data from 84 participants. Using the variance of DCC time-series data as input features, we trained predictive models of the Ruminative Response Scale (RRS) subscales. The "B" stands for the brooding subscale, "D" for the depressive rumination subscale, and "R" for the reflective pondering subscale. We used Lasso regression with leave-one-participant-out cross-validation. We then selected and tested only good-performing models on the next independent test datasets. **b** Among the initial 60, we selected seven predictive models that showed significant cross-validated prediction performance

($q < 0.05$, false discovery rate) in the training dataset ($n = 84$). Among the seven predictive models, we again selected one predictive model that showed significant independent prediction performance at $p < 0.05$ (one-sided permutation test) with the validation dataset ($n = 61$). The selected model was the dmPFC-based predictive model of depressive rumination. We finally tested the model on the last independent test dataset ($n = 48$) to evaluate the model's generalizability. A red-dashed circle indicates the data point that was identified as an outlier (i.e., greater than three standard deviations away from the mean), which did not affect the significance after its removal ($r = 0.276$, $p = 0.028$, one-sided permutation test, 95% CI [−0.012, 0.579]).

**Table 1 | Training and testing results of all models**

| Seeds | Training (n = 84) | | | | | | Test 1 (n = 61) | | | | | | Test 2 (n = 48) | |
|---|---|---|---|---|---|---|---|---|---|---|---|---|---|---|
| | Brood | | Depressive | | Reflective | | Brood | | Depressive | | Reflective | | Depressive | |
| | r | p | r | p | r | p | r | p | r | p | r | p | r | p |
| dMPFC | -0.114 | 0.856 | **0.342** | **0.001** | 0.062 | 0.287 | | | 0.240 | 0.037 | | | 0.288 | 0.025 |
| vMPFC | **0.288** | **0.003** | -0.180 | 0.947 | 0.015 | 0.441 | 0.131 | 0.153 | | | | | | |
| HF (L) | -0.284 | 0.996 | 0.071 | 0.256 | -0.058 | 0.694 | | | | | | | | |
| HF (R) | 0.014 | 0.444 | -0.118 | 0.854 | -0.170 | 0.937 | | | | | | | | |
| LTC (L) | 0.052 | 0.321 | -0.106 | 0.831 | 0.107 | 0.167 | | | | | | | | |
| PCC (L) | **0.392** | **0.000** | 0.042 | 0.356 | -0.046 | 0.670 | -0.109 | 0.795 | | | | | | |
| PHC (L) | 0.195 | 0.037 | 0.096 | 0.188 | -0.008 | 0.531 | | | | | | | | |
| Rsp (L) | -0.116 | 0.854 | 0.289 | 0.006 | 0.259 | 0.009 | | | | | | | | |
| TPJ (L) | 0.023 | 0.433 | -0.007 | 0.520 | 0.221 | 0.019 | | | | | | | | |
| LTC (R) | 0.166 | 0.065 | -0.140 | 0.898 | -0.280 | 0.993 | | | | | | | | |
| PCC (R) | -0.069 | 0.742 | 0.133 | 0.111 | 0.158 | 0.078 | | | | | | | | |
| PHC (R) | 0.007 | 0.483 | 0.123 | 0.131 | **0.450** | **0.000** | | | | | 0.131 | 0.158 | | |
| Rsp (R) | -0.231 | 0.982 | -0.281 | 0.996 | -0.179 | 0.947 | | | | | | | | |
| TPJ (R) | -0.153 | 0.916 | -0.102 | 0.822 | **0.564** | **0.000** | | | | | 0.184 | 0.083 | | |
| pIPL (L) | 0.117 | 0.141 | -0.135 | 0.888 | 0.032 | 0.393 | | | | | | | | |
| TempP (L) | **0.314** | **0.001** | 0.009 | 0.468 | **0.294** | **0.003** | -0.045 | 0.631 | | | -0.022 | 0.568 | | |
| aMPFC (L) | 0.101 | 0.176 | 0.235 | 0.015 | -0.165 | 0.933 | | | | | | | | |
| pIPL (R) | 0.126 | 0.131 | -0.039 | 0.635 | 0.237 | 0.015 | | | | | | | | |
| TempP (R) | -0.067 | 0.722 | -0.121 | 0.857 | -0.114 | 0.854 | | | | | | | | |
| aMPFC (R) | 0.082 | 0.233 | 0.232 | 0.017 | 0.250 | 0.012 | | | | | | | | |

We used 20 regions-of-interest within the default mode network as seeds to create the dynamic functional connectivity features and developed predictive models for three RRS subscales with these features. We started from 60 models (20 × 3) with the training dataset and passed only the models with significant performance to the next round of testing using two additional independent test datasets. Significant correlation values were shown in bold. To obtain unbiased estimates of prediction performance, we used the leave-one-participant-out cross-validation for the training data, and the significance was corrected with the false discovery rate (FDR) $q < 0.05$ for multiple comparisons ($p < 0.006$, for FDR $q < 0.05$). In the test datasets, we tested only significant models from the previous validation procedure. Across the test datasets, only the dmPFC-based depressive rumination predictive model survived. 95% confidence intervals for correlations are reported in Supplementary Table 2.

*L* left, *R* right, *dmPFC* dorsomedial prefrontal cortex, *vmPFC* ventromedial prefrontal cortex, *HF* hippocampal formation, *LTC* lateral temporal cortex, *PCC* posterior cingulate cortex, *PHC* parahippocampal cortex, *Rsp* retrosplenial cortex, *TPJ* temporoparietal junction, *pIPL* posterior inferior parietal lobule, *TempP* temporal pole, *amPFC* anterior medial prefrontal cortex.

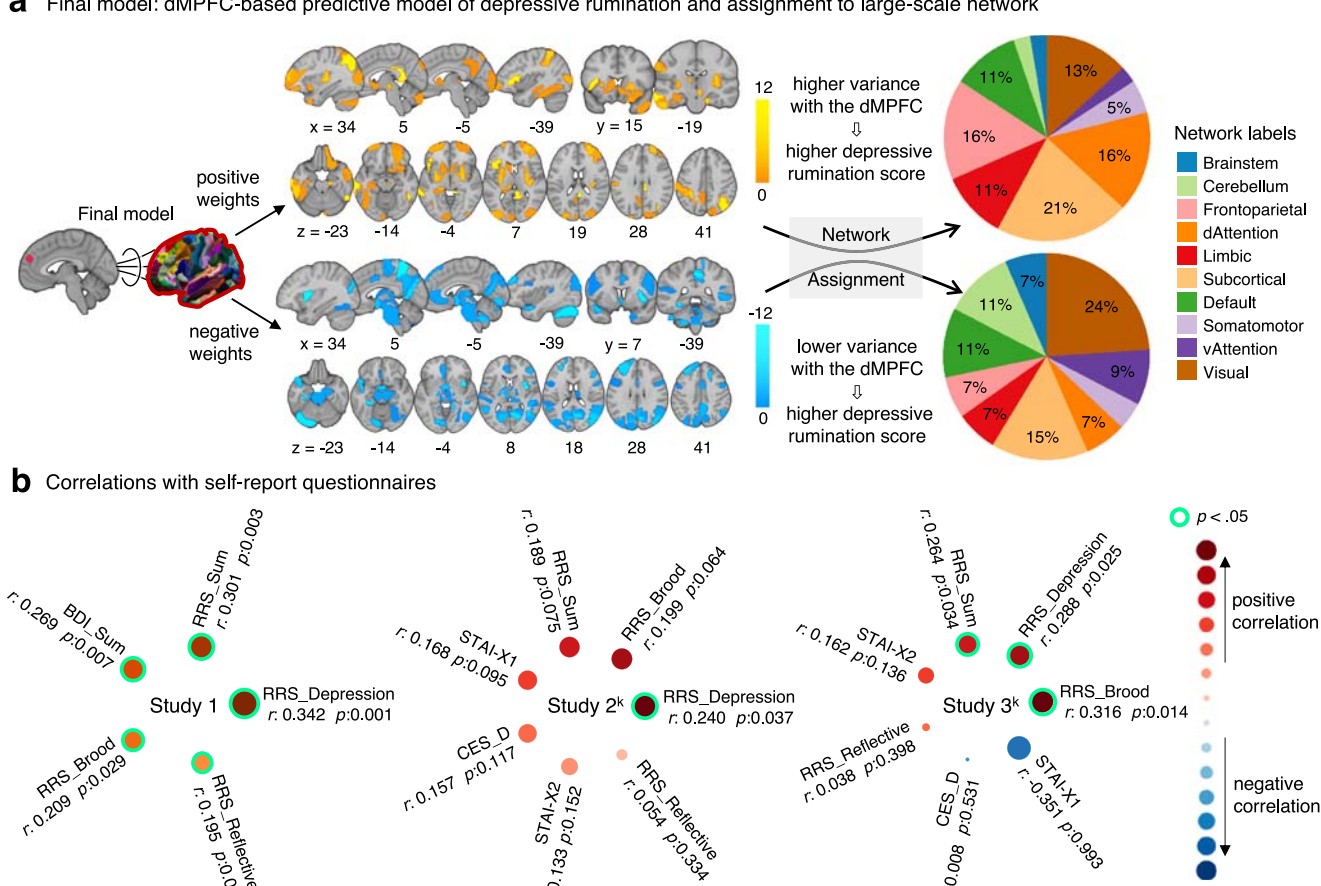

**Fig. 2 | The dmPFC-based predictive model of depressive rumination and its predictions. a** The final model that showed generalizable prediction performances across two independent datasets was the dmPFC-based predictive model of the depressive rumination subscale of the RRS. The top panel shows the brain regions that had positive predictive weights and their large-scale network assignments. Positive weights mean the higher the DCC variances between the dmPFC and the regions, the higher the depressive rumination scores. The bottom panel shows the brain regions that had negative predictive weights and their large-scale network assignments. Negative weights indicate the higher DCC variances between the dmPFC and the regions, the lower the depressive rumination scores. The color bars represent the sign and magnitude of the model weights. The percentage on the pie chart indicates the proportion of region assignments to each large-scale network. A percentage of less than 5% is not shown in the graph. **b** To evaluate the divergent and convergent construct validity of our final model, we examined the correlations between the model prediction and other self-report questionnaires. The superscript k indicates a study that used the Korean translation version for the questionnaires. The questionnaires that showed significant correlations ($p < 0.05$, one-sided permutation test) were marked with green circles.

complex. Among 46 regions with negative weights, 11 regions fell within the visual network, including subregions of the right fusiform gyrus, left parahippocampal gyrus, bilateral lateral occipital complex, and medioventral occipital complex. Seven regions were subcortical areas, including subregions of the bilateral thalamus. Five regions fell within the default mode network, including subregions of the left superior frontal gyrus, right posterior superior temporal sulcus, and right precuneus. And 5 regions were the cerebellum regions, including left cerebellar crus I, left lobule IX, bilateral VIII, left IX, and X. These results highlight that the regions predictive of depressive rumination are distributed across the whole brain rather than confined to one or two specific functional brain networks. To see if our model prediction is also related to other relevant constructs, such as depression and anxiety[37], we examined correlations between our model prediction, i.e., pattern expression, and other self-report questionnaires across datasets. As shown in Fig. 2b, in the training dataset (Study 1), the cross-validated model prediction showed significant correlations with all other RRS subscales and overall RRS summed across subscales, and also with depression measured by BDI. In the validation dataset (Study 2), only the original target variable (i.e., the depressive rumination subscale) showed a significant correlation (green circles in Fig. 2b). In the

second independent test dataset (Study 3), the depressive rumination subscale and the brooding subscale showed significant correlations.

## Identifying regions important for generalization

Next, to examine each region's degree of contribution to the model generalization, we conducted a 'virtual lesion' analysis on two independent test datasets (Fig. 3). In the virtual lesion analysis, we removed one region at a time from the model and calculated the changes in the prediction performance between the full and reduced models. We defined the importance of each region as the difference between the prediction-outcome correlation of the full model versus the reduced model, i.e., $r_{full} - r_{reduced}$. We iterated this procedure for all regions included in the predictive model and for two independent test datasets. Through this iteration, we identified 21 important regions that resulted in decreased prediction performance in both datasets (Fig. 3a). The patterns of the differences in the prediction performance were significantly correlated across two datasets $r = 0.628$ ($p = 1$e-10, two-sided, 95% CI [0.520−0.956]), suggesting the consistency in the virtual lesion analysis results across datasets. Figure 3b shows the average importance across two test datasets. The top three important regions were the left inferior

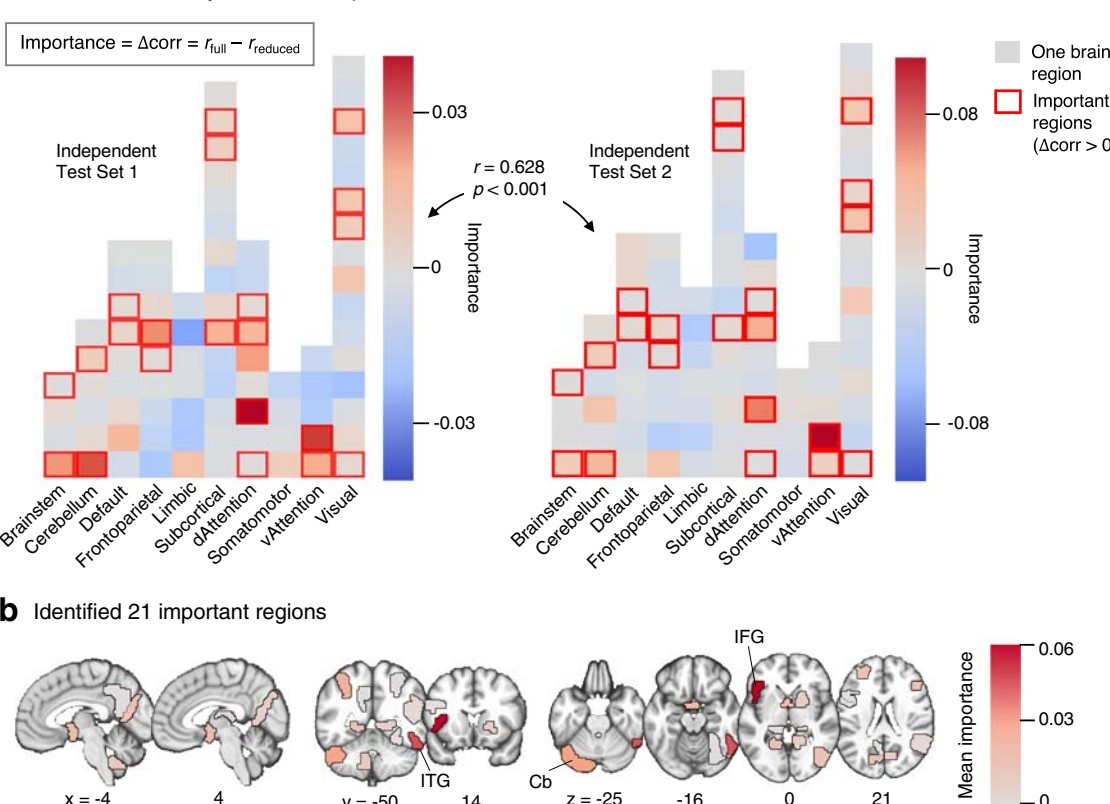

**a** Virtual lesion analysis in the independent test datasets

**b** Identified 21 important regions

**Fig. 3 | Virtual lesion analysis results of the final model. a** Each block represents one brain region out of the 84 regions included in the final model. We grouped them into 10 large-scale functional networks. The color of the blocks indicates the difference in the prediction-outcome correlation between the full model and the reduced model after removing (i.e., virtually lesioning) the brain region from the model. Positive differences indicate that the region is important for the prediction. The boxes with red outlines show the regions with positive difference values across both independent test datasets. There were 21 important brain regions for the prediction. Note that the overall patterns of the region importance were similar across two independent test datasets ($r = 0.628$, $p = 1e$-$10$, two-sided, 95% CI [0.520, 0.956]). **b** The brain map shows the 21 brain regions important for the prediction across two test datasets. Mean difference values across two datasets were shown. The top three most important regions included the left inferior frontal gyrus (lIFG), right inferior temporal gyrus (rITG), and left cerebellum (lCb).

frontal gyrus (IFG), right inferior temporal gyrus (ITG), and left cerebellar crus I.

Figure 4a presents the model only with the identified important regions as a circos plot, in which we visualized the magnitude and sign of predictive weights as line thickness and color, group-level averages of DCC mean connectivity as inner circle boxes, and their large-scale network assignments as outer circle boxes. In addition, we show the important regions' anatomical locations. In this context, DCC mean connectivity represents the average DCC values across time, serving a similar purpose as static connectivity. It indicates the extent of positive or negative correlation, on average, between a region and the dmPFC throughout the resting-state scan. Thus, regions such as the right IPL that exhibit negative predictive weights (blue line in Fig. 4a) and strongly positive mean connectivity (orange inner ring segment in Fig. 4a) exhibited more positive and sustained correlations with the dmPFC over the rest period in ruminative individuals. In contrast, regions such as the left IFG (IFG_L_6_5 A44op according to the Brainnetome nomenclature) that have positive predictive weights and positive mean connectivity have more variability in their correlations over time in ruminative individuals. In terms of the large-scale functional networks, the 21 important regions were distributed across multiple functional networks except for the limbic and somatomotor networks, which contained no important regions. Rumination was consistently associated with more variable dmPFC connectivity with the left IFG (in the ventral attention network), right ITG, striatum, and superior parietal regions, and less variable connectivity between the

dmPFC and the cerebellar and brainstem regions. In addition, Fig. 4b shows that there was no apparent relationship between the predictive weights and the DCC mean connectivity values, suggesting that the relationship between the DCC variance and rumination scores was not dependent on the connectivity magnitude itself. Examining the DMN regions also supported this finding that all the DMN regions showed positive DCC mean values with the dmPFC, but both positive and negative predictive weights were found in these regions (Supplementary Fig. 1).

**Testing the model on individuals with MDD**

To see if our model could predict individual differences in depressive symptoms of individuals clinically diagnosed with MDD, we tested our model on a clinical dataset with 35 people diagnosed with MDD. We tested both the full model, which included all 84 regions, and the refined model, which included only 21 regions identified to be important for model generalization. We found that the full model was not predictive of the BDI-II scores of individuals with MDD ($r = 0.150$, $p = 0.391$, one-sided permutation test, 95% CI [−0.195, 0.498]), whereas the refined model consisting of 21 important regions showed a significant prediction of the BDI-II scores of $r = 0.431$ ($p = 0.010$, one-sided permutation test, 95% CI [0.115, 0.808]) as in Fig. 4c. However, it should be noted that our model fails to generalize when applied to the additional three datasets of patients with MDD ($n = 21$, 57, and 22; Supplementary Fig. 5). The primary distinction between the datasets in which our model worked and those in which it did not included

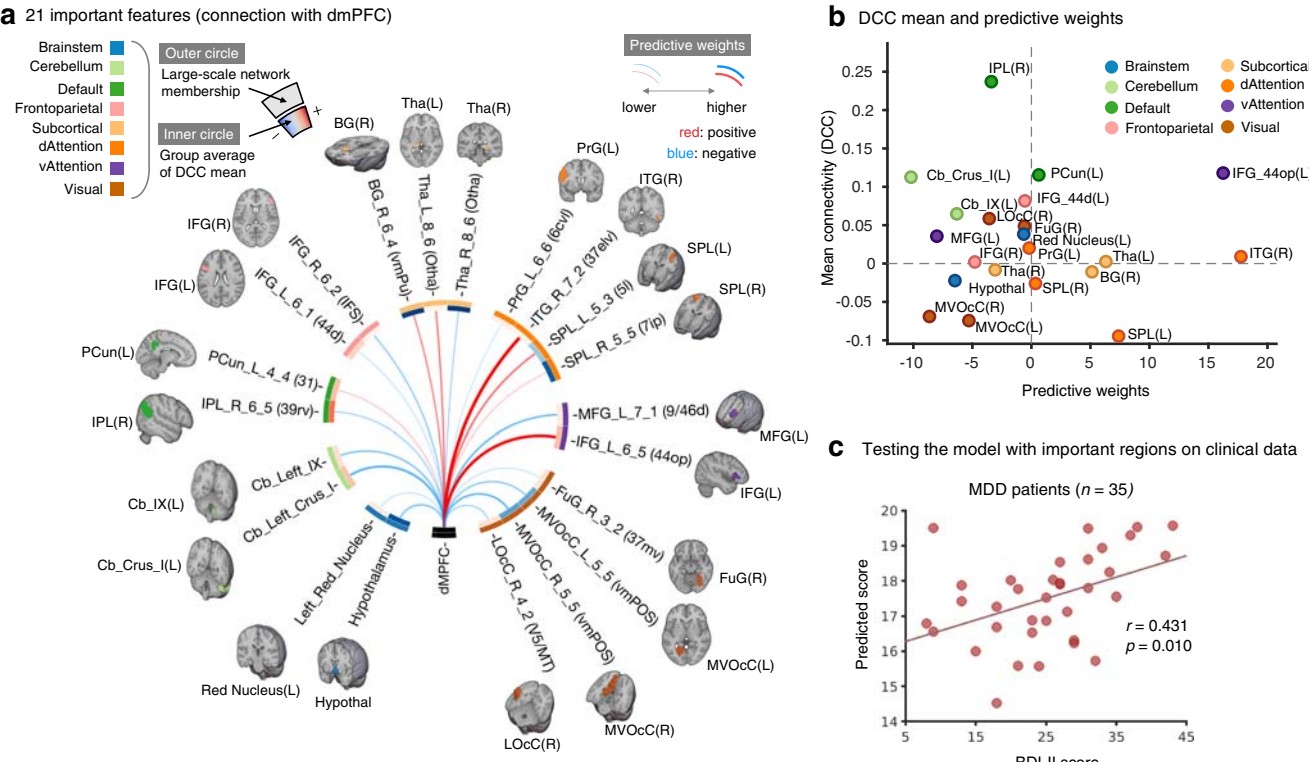

**Fig. 4 | Twenty-one important features of the predictive model. a** Circos plot showing 21 important regions identified with the virtual lesion analysis. The region names outside of the circos plot were from the Brainnetome nomenclature. The outer circle represents the color coding of large-scale networks, while the inner circle represents the mean values of DCC connectivity (equivalent to static connectivity). The line color inside the circos plot indicates the sign of the predictive weights based on DCC temporal variance (i.e., red: positive, blue: negative), while the line thickness indicates the magnitude of the predictive weights. **b** Relationship between the predictive weights and the DCC mean connectivity values of 21 important regions. **c** The scatter plot shows the testing results of the model on Study 4 clinical data from people with MDD ($n = 35$). It shows a significant correlation between the model prediction and the BDI-II score of $r = 0.431$ ($p = 0.010$, one-sided permutation test, 95% CI [0.115, 0.808]).

the phase encoding direction and MRI manufacturer, which were the most significant contributors to the measurement bias according to Yamashita et al.[38].

## Discussion

In this study, we developed a predictive model of depressive rumination based on the temporal dynamic features of functional connectivity using DMN regions as seeds. Through comprehensive tests for model generalizability across three independent datasets (total $n = 193$), we identified a generalizable predictive model of depressive rumination based on the temporal variance of dynamic connectivity between the dmPFC and brain regions distributed across multiple brain systems. Through a virtual lesion analysis across two test datasets, we identified 21 regions important for model generalization, including the left IFG, right ITG, right IPL, cerebellum, and others. Finally, this 21-region model significantly predicted depression scores in individuals ($n = 35$) diagnosed with MDD.

From our extensive search for generalizable predictive models of the three RRS subscales, only the dmPFC-based predictive model for the depressive rumination subscale showed significant prediction performance across all three independent datasets. Considering that ruminative thinking involves dwelling on thoughts that are often high-level, negative, self-referential, past-focused, and verbally-mediated[30,39,40], our findings support our hypothesis that the DMN's dorsal medial subsystem plays an important role in a high-level "mind's mind" form of imagination[21]. This interpretation is further supported by our findings of other key regions within a broader dmPFC subsystem that dynamically interact with the dmPFC in our predictive model, including right IPL and left IFG (see below for further

discussion). Previous studies also showed distinct fMRI patterns of the dmPFC in individuals with MDD compared to healthy controls[16,22,31]. Our study builds upon these findings by developing a generalizable predictive model, providing a deeper understanding of the role of the dmPFC in rumination among individuals with both subclinical and clinical depression.

Beyond the regional dynamics, the importance of network-level dynamics in rumination has been suggested in previous studies[41,42]. For example, Karapanagiotidis et al.[41] showed a close relationship between trait negative affectivity (e.g., anxiety, depression, and rumination) and the DMN-dominant states identified by hidden Markov modeling. In addition, Goodman et al.[42] reported that the dwell time and state transition frequency of DMN-dominant states identified with the coactivation pattern analysis were correlated with depressive symptoms. Converging evidence from the studies that used different analysis methods highlights the importance of the dynamics of the DMN in rumination. In addition to the DMN, our study found that some brain regions outside of the DMN, such as the left IFG from the ventral attention network, are also crucial for rumination. Thus, future studies examining how across-network dynamics are related to rumination would be required for a better understanding of neural dynamics underlying the ruminative thinking.

Even though the predictive features of our model were largely distributed across the whole brain, several regions stand out as particularly important for model generalization, including the left IFG (IFG_L_6_5 according to the Brainnetome nomenclature), right ITG (ITG_R_7_2), left cerebellar crus I, and right IPL (IPL_R_6_5), which all had positive mean connectivity with the dmPFC. Among these, the left IFG appeared to be the most important for generalization across two

datasets. This region is part of the dorsal medial subsystem at rest[19,43] and is known for its importance in language processing[44,45], potentially indicating that rumination predominantly has a verbal representation, which is also associated with a mind's mind form of cognition. This is also consistent with Vatansever et al.[46], which used the canonical correlation analysis to simultaneously decompose resting-state fMRI connectivity and behavioral components. They reported that the brain component that comprised left IFG and a part of the dorsal medial subsystem was related to verbal, negative, and deliberate thoughts. In addition, both the left IFG and dmPFC were identified to be part of the core emotional appraisal system[47], and particularly, the left IFG was shown to play a dual role in emotion generation and regulation[48]. Note that the identified left IFG showed some overlap with the salience network, which could be relevant to a recent finding that higher salience network flexibility was linked to higher negative repetitive thoughts[49].

The right IPL is another region demonstrating positive mean connectivity with the dmPFC, but this time with lower temporal variability related to higher trait rumination. The region labeled "right IPL" in the Brainnetome atlas overlaps with a region commonly referred to as the right temporoparietal junction (TPJ), which has been reported across numerous fMRI studies of mentalizing or theory of mind[50,51]. As a key node of the dorsal medial subsystem, sustained positive correlations between the dmPFC and the right IPL/TPJ at rest may signify the presence of social inferences and evaluations associated with rumination at rest, especially in relation to the self. In addition, the right IPL/TPJ has also been identified as a key area for affective appraisal[52,53], and the region sits at a "convergence zone" along the cortical hierarchy[54], supporting its relevance to integrative, high-level conceptual processing. These suggest that sustained, positive connectivity between the right IPL/TPJ and the dmPFC may reflect the sustained focus of one's affective states, which is a key feature of rumination.

On the left cerebellar crus I of our model, more than 50% of the voxels within the region were within the frontoparietal network[55]. Other important cerebral regions within the frontoparietal network (IFG_R_6_2 and IFG_L_6_1 in Fig. 4a) also showed the same characteristics in predictive weights and DCC mean (i.e., negative predictive weights and positive DCC mean) as the cerebellar crus I. This suggests a possibility that the stable between-network connectivity, particularly between the default mode and frontoparietal networks, underlies the ruminative cognitive process[56,57]. In addition, the right ITG and the left cerebellar crus I were within one of the functional connectivity-based clusters identified by Eickhoff et al.[58], who divided the functional network of the dmPFC into four clusters. The cluster relevant to our finding was the caudal-right cluster, which was strongly connected to the frontoparietal and dorsal attention networks. This may imply that our right ITG and the left cerebellum findings, also with the opposite sign of the weights, would be related to the disrupted adaptive attention control that may be important for rumination.

Interestingly, regions within the visual cortex had negative weights in our model, indicating that more stable connections between the dmPFC and visual areas are predictive of a higher level of rumination. Considering together with the fact that the visual cortex regions showed weak mean functional connectivity with the dmPFC (Fig. 4b), this negative predictive weight may reflect the tendency of diverting one's attention away from perception, also known as perceptual decoupling[59,60], to be predictive of rumination. This is consistent with a recent study[61], in which the DMN activity related to internally oriented cognition was decoupled with the activity in visual cortex regions at rest or during tasks. In addition, the decoupling was enhanced when participants were engaged in autobiographical memory recall.

In addition to the RRS depressive rumination subscale, our model also showed a significant prediction for the BDI-II score in patients with MDD. This finding suggests the presence of a neural pattern that accounts for a continuum between ruminative tendencies in the healthy population and depressive symptoms in the clinical population. Notably, depressive rumination items in the RRS include some similar items to the BDI-II (e.g., "think about how sad you feel" in the RRS and "I feel sad" in the BDI-II). For this reason, the depressive rumination subscales have been suggested to be confounded with depression[62,63]. However, our model did not predict the CES-D score in healthy participants, suggesting that our brain model captures subclinical ruminative tendencies but not depression in healthy participants. These findings highlight the need for future studies to administer the same trait and symptom questionnaires to both subclinical and clinical groups. Additionally, longitudinal assessments of symptoms and brain-based measures would enable tracking of within-person changes over time.

Our study has some limitations that should be addressed in future studies. First, the current study aimed to balance sparsity with complexity by focusing on multiple seed regions across the DMN, by permitting target regions across the entire brain, and by employing Lasso regression to build predictive models of rumination. While acknowledging that restricting the search space to DMN seeds may increase selection and confirmation bias in our findings, this approach capitalizes on a substantial body of research in humans and animals that highlights the involvement of the DMN in emotion and rumination processes. Moreover, this approach facilitates the interpretation and the integration of our findings with prior literature, which would be challenging with models allowing for increased complexity. Our approach, which allows for both hypothesis-driven and exploratory findings, is also advantageous in its ability to increase power, reduce false positives, and balance type I and type II errors. However, future studies, especially if well-powered, could benefit from a more exploratory approach (e.g., combining multiple models, including more features, using less sparse models, etc.).

Second, recent studies suggest that a small amount of rsfMRI scan data (less than 10 min) could result in poor reliability of functional connectivity measures at the individual difference level[64–67]. Our studies used 6–10 min of rsfMRI data, and future studies should address the reliability of our model and the potential benefits of longer scans.

Third, our prediction results showed only small effect sizes, and our sample sizes were small for predicting individual differences[68], suggesting a possibility of type I errors in this study. To further test the generalizability of our model, we tested our model on an additional subclinical independent dataset ($n = 60$; see captions of Supplementary Fig. 6 for the details of this dataset) that we finished collecting during revision. This Supplementary dataset had an interesting experimental design feature: we administered two 14-min resting-state runs both before and after participants viewed a short emotional movie lasting approximately 10 mins. This movie was about a mother meeting her daughter who passed away through virtual reality, and we selected this movie to enhance internally oriented cognitive and emotional states. As shown in Supplementary Fig. 6, our model showed a significant prediction of depressive rumination only with the post-movie resting-state data ($r = 0.228$, $p = 0.040$, one-sided permutation test, 95% CI [−0.028, 0.492]). With the pre-movie resting-state data, our model showed a non-significant prediction ($r = −0.038$, $p = 0.613$, one-sided permutation test, 95% CI [−0.298, 0.222]). Again, the results showed a small effect size and one negative result, which could still suggest a possibility of type I error. However, this also provides an interesting hypothesis that inducing a ruminative cognitive state would increase the prediction performance of our model, which should be examined in future studies. Overall, though we cannot provide definitive evidence that our findings are not false positives in the current study, we put our best effort into further testing our model, and here we provide one positive and one negative result, with an interesting hypothesis for future study.

Fourth, it is important to note that high temporal variance implies large fluctuations over time, but it does not measure how 'frequent' or how 'fast' connectivity changes. Thus, our model might not be fully utilizing the temporal dynamic information, and future studies may benefit from characterizing the temporal frequencies and using more fine-grained temporal information for predictive modeling of rumination, such as frequency[42] and coherence measures[69]. In addition, larger sample sizes and signals with higher temporal precision (e.g., magnetoencephalography) may be necessary for adequate inference for specific frequency bands.

Fifth, our study focused on predicting trait rumination rather than state rumination. The RRS scores used in our study, therefore, do not necessarily capture individuals' rumination during the actual period of scanning, although individuals with high trait rumination may be more likely to experience thoughts characteristic of state ruminative thinking at rest[30,70]. Future studies should target both types of rumination, which may be characterized by different neural mechanisms[71] or show different patterns of connectivity[72].

Sixth, the model's generalizability was affected by which datasets were used in the training procedure (Supplementary Tables 4, 5). Our additional analyses suggested that it could be due to (1) differences in the distribution of the outcome variables across datasets (Supplementary Fig. 3) or (2) the minimum number of input features required for the generalizable prediction of rumination (Supplementary Fig. 4, Supplementary Tables 6, 7). However, it is difficult to know whether there is a specific number of features required for successful prediction. Also, it is possible that multiple modeling options, such as input features (e.g., connectivity vs. activity), resolution (e.g., voxel-level vs. region-level), etc., interact with the required number of features. Future studies should examine these influences in more detail.

Seventh, our study included only a small sample size of individuals with MDD ($n = 35$), and therefore, our model performance in predicting depression in populations with MDD should be further validated. We also showed that our model does not generalize to the clinical datasets with different scan parameters, such as phase encoding direction and MRI manufacturer[38]. In addition, we have not considered specific details regarding depression status (e.g., first episode depression or recurrent depression) nor medication status (e.g., drug-naive or not), which could potentially lead to distinct connectivity profiles within DMN[73]. Also, we could not directly test our model to predict rumination in patients with MDD because the clinical datasets did not have the RRS scores, making it difficult to know what exactly our model predicted in the clinical sample. Thus, future studies should investigate the boundary conditions under which our model works or does not work.

Lastly, our study cannot pinpoint which cognitive functions are supported by the dmPFC due to the limitation of resting-state fMRI and our lack of measurement of concurrent thought processes. In future studies, researchers should use task-based fMRI to specify which types or contents of ongoing thoughts are linked to the region's dynamics. Turnbull et al.,[74] provides a relevant example, in which they demonstrated that the left dorsolateral prefrontal cortex prioritizes individuals' internal thoughts when a situation is non-demanding by showing the region's different connectivity profiles with DMN when the task requires high or low demands. Such specification of the single region will be important for enhancing our understanding of cognitive processes related to rumination.

Despite these limitations, the study provides meaningful progress beyond previous literature by developing an integrated, predictive brain model of rumination that generalizes across multiple samples, including a clinical population, without any readjustment of the model. Recent work suggests that such models have much larger effect sizes[75] and reliability[76] than isolated features. In addition, notably, the model was based on dynamic connectivity features, suggesting that meaningful information is contained in the variability of brain connectivity over time. Our findings will facilitate future directions aiming at a mechanistic understanding of how the dmPFC and its interactions with other regions increase or decrease rumination, helping develop new therapeutic strategies for depression and anxiety.

## Methods

### Participants

Study 1 consisted of 110 healthy adults recruited across the greater Denver area and from existing subject databases assembled by the University of Colorado Boulder Institute for Behavioral Genetics. The sample was designed to oversample non-Hispanic Black American and Hispanic White American participants, to include equal proportions of non-Hispanic African American, Hispanic White American, and Non-Hispanic White American participants. We excluded data from 26 participants due to missing data in either rumination scores or rsfMRI, resulting in a final sample size of $n = 84$ (age = 28.0 ± 4.9 [mean ± SD], 41 male; see Supplementary Table 1 for the demographic and self-report summary information). The final sample included 34 Non-Hispanic White Americans (19 female), 30 Hispanic White Americans (15 female), and 20 Non-Hispanic African American participants (9 female). Most participants also underwent a separate, unrelated experiment[77]. We excluded participants with MR-related contra-indications, psychoactive or pain medications, current or recent (past 6 months) neurological or psychiatric diagnoses, or pain-related medical conditions. The institutional review board of the University of Colorado Boulder approved the study. All participants provided written informed consent and were compensated in cash for their participation.

Studies 2, 3 (validation and independent test datasets), and the Supplementary dataset included 61 (age = 22.9 ± 2.5 [mean ± SD], 31 males), 48 (age = 22.8 ± 2.4, 28 males), and 60 healthy participants (age = 23.35 ± 1.91, 30 males), respectively. We recruited the participants from Suwon, South Korea. The institutional review board of Sungkyunkwan University approved the studies. All participants provided written informed consent and were compensated in cash for their participation. The preliminary eligibility of participants was determined through an online survey. Participants with psychiatric, neurological, or systemic disorders and MRI contraindications were excluded.

Study 4 (an independent clinical test dataset) included 35 individuals diagnosed with MDD who had both rsfMRI data and BDI-II score (age = 44.08 ± 12.1, 18 male). We obtained these clinical data from a publicly available database, the Strategic Research Program of Brain Sciences (SRPBS) Multi-disorder MRI Database[38,78]. This SRPBS database included rsfMRI data of people diagnosed with multiple psychiatric disorders collected from 11 sites. All experimental protocols in the datasets were approved by the institutional review boards of the principal investigators' respective institutions (Advanced Telecommunications Research Institute International (ATR), Hiroshima University, Kyoto Prefectural University of Medicine, Showa University, and the University of Tokyo). All participants in the studies provided written informed consent. We only used data collected from the Center of Innovation at Hiroshima University because it had the largest number of individuals with MDD ($n = 71$). Among them, we excluded participants who were left-handed ($n = 2$) and showed mean frame-wise displacement over 0.25 ($n = 34$), resulting in a total of $n = 35$.

No statistical method was used to predetermine the sample size. Reported sex and/or gender were based on self-reports of participants, and the ratio was made as evenly as possible to make a generalizable prediction across sex and/or gender.

### Self-report questionnaires

**Ruminative response scale (RRS[34]).** The RRS is a self-report measure of how much individuals repetitively, and persistently dwell on their own internal states. It consists of 22 items rated from 1 (never) to 4 (always) and subdivided into three subscales—brooding, reflective

pondering, and depressive rumination[34,79]. There have been some debates on which factors comprise a reliable construct of rumination[80–82]. Here, we used the sum of each subscale of RRS suggested in ref. 79 as a measure of each subscale construct. Treynor et al. argued that the depressive rumination subscale might be confounded with depression itself, while the other two are unconfounded with depression and each reflects adaptive (reflective pondering) and maladaptive (brooding) aspects of rumination. We used the subscale scores as dependent variables in our model development (Fig. 1). Studies 2–3 used the Korean version of RRS[83], which included only 19 items (Item numbers 2, 14, and 15 from the original scale were excluded) and had a slightly different factor structure compared to the original scale. The brooding subscale was comprised of 6 items (Item numbers 5, 9, 10, 13, 16, and 18), reflective pondering was comprised of six items (Item numbers 7, 11, 12, 20, 21, and 22), and depressive rumination was comprised of seven items (Item numbers 1, 3, 4, 6, 8, 17, and 19).

**Beck Depression Inventory (BDI[84,85]).** The BDI is a 21-item self-report questionnaire that assesses individuals' depression severity[84]. Each item rating ranges from 0 to 3 with increasing severity of depressive symptoms. BDI-II is a revised version of the BDI[85]. BDI-II was administered in Study 1 and Study 4.

**Center for Epidemiological Studies-Depression (CES-D[86]).** The CES-D is a 20-item self-report questionnaire that assesses individuals' depression severity. Each item ranges from 0 (Rarely or none) to 3 (Most or almost all the time). We used the sum score after reverse-scoring positively keyed items (Item numbers 4, 8, 12, 16). We administered the Korean translation of CES-D in Studies 2 and 3.

**State-Trait Anxiety Inventory-X form (STAI-X[87]).** STAI-X is a self-report questionnaire that assesses the levels of state and trait anxiety of the individuals. It consists of 20 state anxiety items (STAI-X1) and 20 trait anxiety items (STAI-X2). Both types of items range from 1 (not at all or never) to 4 (very much so or always). We used the sum score of STAI-X1 and STAI-X2 after reverse-scoring positively keyed items (Item number 1, 2, 5, 8, 10, 11, 15, 16, 19, 20 for STAI-X1; 1, 6, 7, 10, 13, 16, 19 for STAI-X2). We administered the Korean translation of STAI-X items in Studies 2 and 3. See Supplementary Fig. 2 for the correlations among the self-report questionnaires.

**Resting-state fMRI Paradigm**
**Study 1 (Training dataset).** Participants in Study 1 were asked to stare at a centrally positioned fixation crosshair for 7 min and let their thoughts flow naturally. This resting-state paradigm was followed by additional tasks relevant to other studies (e.g., ref. 77). Note that the Study 1 resting-state data were collected from the same participants as in ref. 77, which did not use the resting-state data.

**Study 2 and 3 (Validation and testing datasets).** Resting-state runs in Studies 2 and 3 were also part of larger studies. Six minutes of resting-state data were acquired while participants fixated on a central point displayed in the center of the screen. Psychtoolbox (version 3.0, http://psychtoolbox.org) was used to display the fixation cross. In Study 3, during the resting scan, we intermittently asked participants to report their momentary thought content using a word or phrase. We conducted this thought sampling 5 times during the run while the interval between questions was around 1 min. We regressed out thought sampling-related fMRI signal (for details on the nuisance regression, see the next section).

**Study 4 (Independent clinical test dataset).** Study 4 acquired 10 min of resting-state fMRI data while participants were asked to look at a fixation point on the screen. More details are in ref. 38.

**fMRI data acquisition and preprocessing**
Study 1 data were acquired on a 3 Tesla Siemens Trio MRI scanner located in the Center for Innovation and Creativity at the University of Colorado Boulder. High-resolution T1-weighted magnetization-prepared rapid gradient echo (MPRAGE) images were acquired with TR = 2530 ms, TE = 1.64 ms, flip angle = 7°, inversion time (TI): 1200 ms, field of view (FoV) read: 256 mm, echo spacing: 12.2 ms, bandwidth: 651 Hz Px−1, time: 6:03 for normalization. For functional EPI images, a multi-band sequence was used with TR = 460 ms, TE = 29 ms, slices = 56, multi-band factor = 8, flip angle = 44°, FoV read = 248 mm, echo spacing = 0.51 ms, bandwidth = 2772 Hz Px−1, time = 10:15. Then, T1-weighted MPRAGE images were co-registered to the mean functional image and normalized to the MNI-152 template using SPM8. For functional EPI images, the initial images of every functional scan were removed for image-intensity stabilization. The remaining images were realigned to the first image to correct for head motion using SPM8, warped to the MNI-152 template using warping parameter from co-registration of the T1-weighted image, interpolated to $2 \times 2 \times 2$ mm$^3$, and finally smoothed with an 8 mm FWHM Gaussian kernel. Then, temporal data were band-pass filtered to include temporal frequencies between 0.008 Hz and 0.1 Hz. Nuisance covariates included image-intensity outliers (i.e., "spikes"), 24 head motion parameters (six movement parameters including x, y, z, roll, pitch, and yaw, their mean-centered squares, their derivatives, and squared derivative), and the top five component scores from each of the white and cerebrospinal fluid (CSF) masks. To remove intermittent gradient and severe motion-related artefacts present to some degree in all fMRI data[88], spikes were identified by computing the Mahalanobis distances for the matrix of slice-wise mean and standard deviation values by functional volumes, and also by calculating root-mean-squared successive differences across images.

Studies 2 and 3 were acquired on a 3 T Siemens Prisma scanner located in the Center for Neuroscience Imaging Research at Sung-kyunkwan University. High-resolution T1-weighted structural images were acquired. Functional EPI images were acquired with TR = 460 ms, TE = 27.2 ms, field of view = 220 mm, $82 \times 82$ matrix, $2.7 \times 2.7 \times 2.7$ mm$^3$ voxels, 56 interleaved slices, 2608 volumes. Preprocessing of structural and functional images was performed with SPM12 (Wellcome Trust Centre for Neuroimaging, London, UK) and FSL. For structural T1-weighted images, magnetic field bias was corrected, and non-brain tissues were removed (skull-stripping). Then, the images were normalized to MNI space. For functional EPI images, initial volumes (20 images) of fMRI images were removed to allow for image-intensity stabilization. Then, the images were distortion-corrected, motion-corrected (realigned), co-registered with T1-weighted images, normalized to MNI with the interpolation to $2 \times 2 \times 2$ mm$^3$ voxels, and smoothed with a 5 mm FWHM Gaussian kernel. The remaining preprocessing techniques, including band-pass filtering and regressing out of nuisance covariates were the same procedures as outlined in Study 1. The Study 3 dataset included additional nuisance covariates corresponding to visual cues asking for words, and reporting time to regress out cue and speaking effects.

Study 4 data were acquired through a 3 T Siemens Magnetom Verio scanner. High-resolution T1-weighted structural images were acquired. Functional EPI images were acquired with TR = 2500 ms, TE = 30 ms, field of view = 212 mm, $64 \times 64$ matrix, $3.3 \times 3.3 \times 3.2$ mm$^3$ voxels, and 40 ascending slices. All the preprocessing steps were the same as in Studies 2 and 3 except that there was additional slice timing correction using SPM12 and no distortion correction.

**Developing a dynamic connectivity marker for rumination**
We used the variance of seed-based dynamic conditional correlations (DCC)[36] as an input feature for developing predictive brain models of rumination. The DCC is a model-based dynamic

correlation estimation method using a combination of generalized autoregressive conditional heteroscedastic (GARCH) and exponential weighted moving average (EWMA) models. We chose to use the DCC based on results from ref. 36, which tested multiple dynamic functional connectivity methods, including the sliding-time windows method with various window sizes and reported that the DCC showed a good balance between the sensitivity and specificity in estimating the variance of dynamic functional connectivity. In addition, the DCC is known to have a higher level of test-retest reliability in estimating the variance[89] and does not require any additional arbitrary hyperparameters. As seed regions, we used 20 DMN subregions from ref. 35, including the left and right hemispheres separately, except for the medial regions (i.e., the dorsal and ventral medial prefrontal cortex regions). For the whole-brain data, we used a modified version of the Brainnetome parcellation[90], which included the original 246 Brainnetome parcels and 34 additional parcels for midbrain, brainstem, and cerebellar regions[91,92], resulting in a total of 280 regions. As outcome variables, we used participants' scores from the three subscales of the RRS—depressive rumination, brooding, and reflective pondering.

Using the Study 1 dataset ($n = 84$), we trained a total of 60 models, i.e., 20 seed-based features (DCC variance) × 3 outcome variables (RRS subscales), with the least absolute shrinkage and selection operator (Lasso) regression. Using Lasso regularization, we selected the maximum number of features, i.e., the sample size, $n = 84$. We chose to select the maximum number of features to avoid additional hyper-parameter searches, which could cause overfitting. The significance of the model performance was evaluated with the correlation between predicted outcomes and actual RRS subscale scores using leave-one-participant-out cross-validation. We used one-sided permutation tests for the model significance testing as our null hypothesis is that the predicted score from the model is not predictive of the actual score, resulting in a correlation similar to or below zero. In addition, permutation tests are non-parametric and only require an assumption of exchangeable variables (across subjects in our case), which fits our case of individual trait prediction. We conducted permutation tests by shuffling participant labels for the actual and predicted RRS scores with 10,000 iterations. We used Pearson correlation as a measure of model performance, instead of squared error-based metrics (e.g., $R$-squared or mean squared error)[93], because the scales of both the fMRI and behavioral data were different across studies and the correlation-based measures are insensitive to the scale difference. All values noted as $r$ in the manuscript indicate Pearson correlations. We selected models that showed significant prediction performance at the false discovery rate $q < 0.05$ and tested them on Study 2 data. From this validation process, we further narrowed our selection by selecting the model that showed significant prediction performance on the Study 2 dataset and testing the selected model on additional independent test data (Study 3). Our decision to use Study 1 as the training dataset was made a priori because it had the largest sample size ($n = 84$), the most racial-ethnic diversity (see "Participants" section.), and the largest variance in trait rumination scores across participants (Supplementary Fig. 3), all of which can help improve the generalizability of our model.

### Model weight assignments to functional resting-state networks

For functional interpretation of the model, we assigned model weights to ten functional groups, including seven networks from ref. 43, subcortical regions[90], brainstem[92], and cerebellum[91]. (Fig. 2a, right). The assignment of cortical regions to seven networks by Yeo et al. was based on network information from each parcellation provided by the Brainnetome Atlas (http://atlas.brainnetome.org/).

### Virtual lesion analysis

To examine the importance of each feature in the prediction, we tested the model performance by removing one feature at a time (i.e., virtual lesion) during independent tests and compared the model performance for the full model versus the reduced model[94,95].

$$\Delta \text{corr} = r_{\text{full}} - r_{\text{reduced}}$$

If the prediction performance decreased after removing a feature (i.e., if $\Delta$corr was positive), we considered the feature to be important for prediction.

### Reporting summary

Further information on research design is available in the Nature Portfolio Reporting Summary linked to this article.

### Data availability

Data from Studies 1–3 and the Supplementary dataset to generate the main results and Supplementary information of the study is provided at the following link (https://github.com/cocoanlab/rumination; https://doi.org/10.5281/zenodo.7923949). Raw data for Studies 1–3 and the Supplementary dataset are available upon request. This is because some participants agreed to share their data only for the limited purpose of scientific research. Any identified research group can request and utilize the shared data for research. To make a request, please send an email to either didch1789@gmail.com (J.K.) or wani-woo@g.skku.edu (C.-W.W.). We will respond to the requests within a week. The data for Study 4 are available at https://bicr-resource.atr.jp/srpbs1600/ (ref. 38).

### Code availability

The codes and data to generate the main figures and results are provided at https://github.com/cocoanlab/rumination and https://doi.org/10.5281/zenodo.7923949. In-house Matlab codes for fMRI data analyses are available at https://github.com/canlab/CanlabCore and https://github.com/cocoanlab/cocoanCORE.

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

## Acknowledgements

This work was supported by IBS-R015-D1 (Institute for Basic Science; to C.-W.W.), 2019R1C1C1004512, 2021M3A9E4080780, and 2021M3E5D2A01022515 (National Research Foundation of Korea; to C.-W.W.), 2E31511-22-090 (KIST Institutional Program; to C.-W.W.), and the

Fourth Stage of Brain Korea 21 Project (to J.K.), and NIH MH076136, DA046064, and R01DA035484 (to T.D.W), and K01DA045735 (to E.A.R.L.).

## Author contributions

J.R.A.-H., H.E., E.A.R.L., C.-W.W., and T.D.W. designed the experiment. B.K.L., H.J.K., E.L., and C.-W.W. contributed to the independent test datasets. J.K. and C.-W.W. analyzed the data. M.A.L. developed the dynamic connectivity algorithm used as part of the analysis. J.K., C.-W.W., and J.R.A.-H. interpreted the results and wrote the manuscript. J.R.A.-H., T.D.W., M.A.L., and C.-W.W. edited the manuscript and provided supervision.

## Competing interests

The authors declare no competing interests.
