## [Peer Review File · Nature Communications]

Reviewers' Comments:

Reviewer #1:

Remarks to the Author:

Review of Woo et al.

I enjoyed reading this paper, it is on a very important topic and the results are interesting, informative and in my opinion appropriate for Nature Communications. An important feature of the paper is the robustness of the findings across data sets, which is an important issue in contemporary work on biomarkers. I am also excited how this work aligns with prior published studies that we have done on very similar topics, which is very important for the development of better neurocognitive accounts of psychiatric difficulties.

My only general comment is that I think it would be appropriate for the authors to include work from my lab (e.g. by Karapanagiotidis et al., (2020) in their revision. In our study we have pursued these questions in a broad manner for several years. For example, Karapanagiotidis et al., applied hidden markov models to resting state data and identified a state dominated by the default mode network, that was related to patterns of negative past related thought during the scan, and predicted higher levels of depression and anxiety. This is very likely a similar phenotype as the authors are studying. It is really encouraging that the two different methods examining neural dynamics can find very similar results and our prior study provides a key indication into the broad experiential landscape within which the current study is embedded within. More broadly when reading the paper I was very excited about the overlap between our more general findings about how internal focus operates and the specific findings here, and including these in an integrated way would be very beneficial to the community since it would help develop a process view of how ongoing thought can be both helpful and costly.

Signed

Jonathan Smallwood

Minor points

Line 206. I thought it was really interesting that the visual cortex has negative weights. We have argued that states of ongoing thought often have this property which we refer to as perceptual decoupling.(Smallwood, 2013). This phenomena may occur in default mode network states when memory dominates experience either when retrieving task relevant material (Murphy et al., 2019, Neuroimage) or when the memories are task irrelevant (Zhang et al., 2022, Elife). It may be worth commenting on this.

Line 337. If I understand this part of the paper correctly, it is really cool that the IFG shows stronger connections with the DMN in rumination here. In one of our prior studies, Vatansever et al., (2016) we found that high IFG connectivity to regions of the DMN was linked to unpleasant verbal thoughts at rest. I don't think we included the dMPFC seed in the analysis, but it is certainly well in line with the idea discussed here and provides important corroboration of both this study, and our prior work. I also found the IFG-saliency network finding. In our work we find that this system can both suppress and facilitate the emergence of patterns of ongoing thoughts with both internal and external features. For example, Turnbull et al., (2018, Neuroimage, 2019, Nature Comms) found dLPFC can suppress and enhance off task thought, and using the ventral attention as a seed region established that connectivity with motor cortex was linked to effectively controlling when off task thought occurs. It may be that the findings here show the other side of the coin (i.e. a pattern linked to worse control).

Line 405. Although our studies go along way to rule out this limitation, I think that it may be appropriate to make it clear that I don't think the authors can distinguish how general the role of dMPFC rumination. It may well be that this regions role in complex thought is much broader than is seen in this study, contributing to multiple types of ongoing thoughts, rumination included. This study is very comprehensive in its goal to map this feature of the dMPFCs contribution to rumination but does not have the capacity to identify other functions (cf the role of the dLPFC we

identified in Turnbull et al., 2019). Please note that in this prior study we also performed an echoes analysis which found that the dorsal medial subsystem of the DMN was likely key to the generation of off task states providing clear overlap with the current findings. In our study, however, this broader system can be linked to better control of thinking.

Reviewer #2:

Remarks to the Author:

The manuscript by Kim et al, titled "A DYNAMIC FUNCTIONAL CONNECTIVITY MARKER FOR RUMINATION" and submitted to the journal Nature Communications, investigates a dynamic functional connectivity predictive model of rumination, a common symptom of depression. The study combines four small datasets to form an unique approach to develop a predictive brain marker. Overall, the study has several strengths, is well written, and shows promising results. The largest weakness in my opinion is that at points the paper feels like it is written in a vacuum, without providing justifications for the choices that they make and how those choices affect downstream interpretation of the results. . I have signed this review for transparency and am happy to discuss these comments if they are unclear. – Dustin Scheinost

A major comment I have is that the authors could better justify their approach and provide comparisons to other approaches. Their approach is novel and appears to perform well but at times details about the exact choices in models were made and comparisons to see if the choices were optimal would be helpful. For example,

Does the dynamic connectivity predict better than static connectivity? I think the authors need to show results based on static connectivity. Otherwise, it is hard to justify starting with dynamic functional connectivity. Alternatively, a strong motivation for dynamic functional connectivity could be presented in the introduction. Have predictive models of static connectivity notably failed in predicting depression or rumination symptoms? The literature is mixed in terms of dynamic functional connectivity being better than static connectivity for predictive modeling and brain behavior associations in general.

Similarly, the choice of Dynamic Conditional Correlation (DCC) should be explained. How is this better suited for the task at hand over the large number of dynamic functional connectivity methods currently available.

While targeting the default mode networks is well justified, why are all seed regions treated independently rather than combining or mix & matching different seeds? Combine all the imaging features together into a single model may result in a more generalizable model.

Relatedly, the authors appear to prefer sparsity in their approach. The lasso model, the validation approach of choosing the "winning" models at each dataset, the focusing on a single seed of the DMN at a time, and the refinement step will all lead to a small number of features being retained in a single model. Which are all fine, but these choices should be discussed. In particular, there is a confirmation bias of sorts in the discussion. The authors chose to look at the DMN because of its role in depression, find that a key node of the DMN is predictive of depression, and then discuss the DMN role in depression. But a different approach that predict similarly well with different brain regions would lead to a different discussion. I guess is this all just to say that I feel that the authors should caution how their modeling choices lead to the studies neurobiological features. Why were each dataset chosen for there specific role (training, testing, validation)? While the training dataset has the largest sample size, the sample sizes of all datasets are of the same order. If the authors iterated through every combination of assigning a dataset to training, testing, and validation, would the same models "win"? Along the same lines, why did the authors take this approach to cross-validation compared to a more standard 10-fold cross validation (even if in the supplement) could be a value to the field.

Finally, why not combine all the behavioral data into a single model? As the predictive model show pretty good cross prediction of results, the brain signatures may not be independent and predicting a latent factor (say from a principal component analysis) of all measures often produces a better prediction results.

I appreciate adding a depression dataset as a final validation. But I worry about the low number of individuals (n = 35) with depression, especially given the heterogeneous nature of depression. There are multiple depression datasets available fully processed and open sources (see REST-meta-MDD Consortium) as well as the other sites in the Strategic Research Program for Brain Sciences dataset.

There appears to be an outlier in independent test sample 2. It likely doesn't affect the results, but the authors may want to check.

A more minor point: I think the paper could increase the diversity of citations as opposed to works from the author's extended group. For example, it could be beneficial to cite: Taxali et al 2021 when discussing that models have greater reliability of than edge level features.

Reviewer #3:

Remarks to the Author:

From this reviewer's perspective, the best aspect of the manuscript is the pattern of dmPFC findings. Across cultures and in the context of assessing the nebulous and ephemeral phenomenon of rumination, it does seem like something about the dmPFC is important. The potential importance of this region as opposed to other DMN structures is presented well in the introduction. In developing the work, the authors might choose to determine if other dmPFC connectivity metrics do or do not predict depressive rumination. If they do, then we might conclude that dmPFC relations with other structures are important in general; if not, then we might reach the even more useful conclusion that connective variability of dmPFC is especially important.

Less good are the biomarker claims made and implied throughout the manuscript. The results are probably consistent but they are not particularly strong, with many parameters coming together to explain 10% of the variance. Indeed, up until the model predicts BDI in the depressed sample, the work smacks of type-I error with seven of 60 models surviving round 1 and one of seven surviving round 2 all the while using lenient FDR corrections and one-sided statistical tests.

The authors state, "We therefore hypothesize that components of the dorsal medial system—especially connectivity with the dmPFC—may be a key neural marker important for both trait and state rumination." How might a region of the brain mark both a cognitive phenomenon in action as well as the tendency for that phenomenon to occur in the absence of that phenomenon? This is important to consider in light of claims that the imaging results might provide "a direct window into depression-relevant brain processes." This is especially noteworthy in consideration of the lack of reported results from the thought-probe resting fMRI approach from Study 3. This scan would seem to provide the only means close to assessing rumination at the state level.

In one sense, the authors' model predicting BDI in a diagnosed depressed sample is quite nice in that the model was trained on rumination of three varieties, only one of which associates strongly with levels of depressive symptoms which, themselves, fluctuate daily in depressed persons. In another sense, it's a stretch to contend that dmPFC DCC with several brain regions is a marker of rumination in depression when rumination in depression was not formally assessed.

To be faithful to the constructs of rumination presented, the authors should make clear that what they are calling "depressive rumination" is rumination that is correlated to levels of depressive symptoms in a depressed cohort studied by Treynor et al. The original notion was to study rumination in depression that was not part and parcel of depressive symptomatology—i.e., brooding and pondering.

In Study 3, the authors "regressed out thought sampling-related fMRI signal." This might give prospective readers the impression that the effects of conducting thought sampling five times during a putative resting-state scan did not otherwise interfere with cognitive and affective processes that occurred during the scan. Thought sampling is a necessary and potentially powerful tool but there is little to do about resulting "Heisenberg effects." In this vein, it is possible that prompting subjects about their thought content caused more internal self-focus than would have otherwise been the case which then biased patterns of resting-brain connectivity toward the authors' theoretical stance.

The concept of reliability is a cornerstone of the authors' study. It is noteworthy, then, that three of the four sub-studies collected less BOLD data than the minimum recommended in order to achieve adequate test-retest reliability of rsfMRI-derived metrics (e.g., <https://www.sciencedirect.com/science/article/pii/S0149763414001262>).

Calculating correlations between changes in delta corr across two studies in the virtual lesion analysis approach is a clever way of showing the statistical reliability of the overall pattern of "lesion" effects. In the end, though, it also shows that there is only ~40% correspondence between models.

Ms. No.: NCOMMS-022-05713

We would like to thank the reviewers for their insightful comments and helpful suggestions and the editors of *Nature Communications* for the opportunity to address these comments. In response to the constructive feedback, we performed additional analyses and revised the manuscript as indicated in the point-by-point responses below.

Summary of main changes:

- ✧ We added four additional supplementary figures (**Supplementary Figs. 3-6**) and three supplementary tables (**Supplementary Tables 2-4**) to address many of Reviewer 2 and 3's comments.
- ✧ We added one additional (non-clinical) test dataset as a supplementary dataset. A brief description of the dataset and the results were added in **Supplementary Fig. 6**.
- ✧ We changed the title of the manuscript to "***A Dorsomedial prefrontal cortex-based dynamic functional connectivity model of rumination***"

We hope our revisions satisfactorily addressed all the issues raised. Again, we appreciate all your insightful comments and the opportunity to improve the manuscript.

(Font color legends: Reviewers' comments are in purple, our responses are in black, and the revisions are in red.)

Reviewer #1

“I enjoyed reading this paper, it is on a very important topic and the results are interesting, informative and in my opinion appropriate for Nature Communications. An important feature of the paper is the robustness of the findings across data sets, which is an important issue in contemporary work on biomarkers. I am also excited how this work aligns with prior published studies that we have done on very similar topics, which is very important for the development of better neurocognitive accounts of psychiatric difficulties.”

Response: We thank the reviewer for the positive evaluation of our paper. We are excited to hear that our study builds on your prior work and hope that such findings could contribute to a better understanding of the brain mechanisms of rumination and spontaneous cognition.

R1-1: *“My only general comment is that I think it would be appropriate for the authors to include work from my lab (e.g. by Karapanagiotidis et al., (2020) in their revision. In our study we have pursued these questions in a broad manner for several years. For example, Karapanagiotidis et al., applied hidden markov models to resting state data and identified a state dominated by the default mode network, that was related to patterns of negative past related thought during the scan, and predicted higher levels of depression and anxiety. This is very likely a similar phenotype as the authors are studying. It is really encouraging that the two different methods examining neural dynamics can find very similar results and our prior study provides a key indication into the broad experiential landscape within which the current study is embedded within. More broadly when reading the paper I was very excited about the overlap between our more general findings about how internal focus operates and the specific findings here, and including these in a integrated way would be very beneficial to the community since it would help develop a process view of how ongoing thought can be both helpful and costly.”*

Response: We appreciate the reviewer’s comment. It was interesting and exciting to see that Karapanagiotidis et al.¹ that used a different analysis method—i.e., the Hidden Markov Model—reported similar findings to our study. In our revision, we tried to integrate Karapanagiotidis et al.¹ to provide a more comprehensive discussion of our findings.

Revision to the main manuscript, p.16 (Discussion):

Beyond the regional dynamics, the importance of network-level dynamics in rumination has been suggested in previous studies^{1,2}. For example, Karapanagiotidis et al.¹ showed a close relationship between trait negative affectivity (e.g., anxiety, depression, and rumination) and the DMN-dominant states identified by hidden Markov modeling. In addition, Goodman et al.² reported that the dwell time and state transition frequency of DMN-dominant states identified with the coactivation pattern analysis were correlated with depressive symptoms. Converging evidence from the studies that used different analysis methods highlights the importance of the dynamics of the DMN in rumination. It also implies a relationship between spatiotemporal characteristics of DMN states (e.g., frequency, dwelling time) and our internal-oriented thoughts beyond rumination. In addition to the DMN, our study found that some brain regions outside of the DMN, such as the left IFG from the ventral attention network, are also crucial for rumination, and thus future studies that show how across-

network dynamics are related to rumination would be required for our better understanding of neural dynamics underlying the process.

R1-2: *“Line 206. I thought it was really interesting that the visual cortex has negative weights. We have argued that states of ongoing thought often have this property which we refer to as perceptual decoupling. (Smallwood, 2013). This phenomena may occur in default mode network states when memory dominates experience either when retrieving task relevant material (Murphy et al., 2019, Neuroimage) or when the memories are task irrelevant (Zhang et al., 2022, Elife). It may be worth commenting on this.”*

Response: We thank the reviewer for the insightful comment on the meaning of the negative weights within the visual cortex regions. We added the following discussion to the manuscript:

Revision to the manuscript, p.18 (Discussion):

*Interestingly, regions within the visual cortex had negative weights in our model, indicating that more stable connections between the dmPFC and visual areas are predictive of a higher level of rumination. Considering together with the fact that the visual cortex regions showed weak mean functional connectivity with the dmPFC (**Fig. 4b**), this negative predictive weight may reflect the tendency of diverting one’s attention away from perception, also known as “perceptual decoupling,”^{3,4} to be predictive of rumination. This is consistent with a recent study⁵, in which the DMN activity related to internally oriented cognition was decoupled with the activity in visual cortex regions at rest or during tasks. In addition, the decoupling was enhanced when participants were engaged in autobiographical memory recall.*

R1-3: *“Line 337. If I understand this part of the paper correctly, it is really cool that the IFG shows stronger connections with the DMN in rumination here. In one of our prior studies, Vatansever et al., (2016) we found that high IFG connectivity to regions of the DMN was linked to unpleasant verbal thoughts at rest. I don’t think we included the dmPFC seed in the analysis, but it is certainly well in line with the idea discussed here and provides important corroboration of both this study, and our prior work. I also found the IFG-saliency network finding. In our work we find that this system can both suppress and facilitate the emergence of patterns of ongoing thoughts with both internal and external features. For example, Turnbull et al., (2018, Neuroimage, 2019, Nature Comms) found dlPFC can suppress and enhance off task thought, and using the ventral attention as a seed region established that connectivity with motor cortex was linked to effectively controlling when off task thought occurs. It may be that the findings here show the other side of the coin (i.e. a pattern linked to worse control).”*

Response: The reviewer points out very interesting connections to his work. In response, we added the following discussion when speculating on the implication of the IFG:

Revision to the manuscript, p.16-17 (Discussion):

This is also consistent with Vatansever et al.⁶, which used the canonical correlation analysis

(CCA) to simultaneously decompose resting-state fMRI connectivity and behavioral components. They reported that the brain component that comprises left IFG and a part of the dorsal medial subsystem was related to verbal, negative, and deliberate thoughts.

R1-4: *“Line 405. Although our studies go along way to rule out this limitation, I think that it may be appropriate to make it clear that I don’t think the authors can distinguish how general the role of dmPFC rumination. It may well be that this regions role in complex thought is much broader than is seen in this study, contributing to multiple types of ongoing thoughts, rumination included. This study is very comprehensive in it’s goal to map this feature of the dmPFCs contribution to rumination but does not have the capacity to identify other functions (cf the role of the dLPFC we identified in Turnbull et al., 2019). Please note that in this prior study we also performed an echoes analysis which found that the dorsal medial subsystem of the DMN was likely key to the generation of off task states providing clear overlap with the current findings. In our study, however, this broader system can be linked to better control of thinking. ”*

Response: We agree with the reviewer that it is difficult to pinpoint which cognitive functions are supported by the dmPFC with our data due to the limitation of resting-state fMRI. Thus, we have added the following text into the discussion:

Revision to the manuscript, p.20 (Discussion) :

Lastly, our study cannot pinpoint which cognitive functions are supported by the dmPFC due to the limitation of resting-state fMRI and our lack of measurement of concurrent thoughts. In future studies, researchers should use task-based fMRI to specify which types or contents of ongoing thoughts are linked to the region's dynamics. Turnbull et al.,⁷ provides a relevant example, in which they demonstrated that the left dorsolateral prefrontal cortex prioritizes individuals’ internal thoughts when a situation is non-demanding by showing the region’s different connectivity profiles with DMN when the task requires high or low demands. Such specification of the single region will be important for enhancing our understanding of cognitive functions.

However, we added new supplementary results (detailed in **R3-2** below) by applying the model to an additional supplementary dataset, which had an interesting experimental design feature—we administered two 14-min resting-state runs before and after participants watched a short emotional movie (around 10 mins long). This movie was about a mother meeting her daughter who passed away through virtual reality, and we selected this movie to enhance internally oriented cognitive states. As shown in **Supplementary Figure 6** (newly added in this revision), our model showed a significant prediction of depressive rumination only with the post-movie resting-state data, $r = 0.228$, $p = 0.040$. With the pre-movie resting-state data, our model showed non-significant prediction, $r = -0.038$, $p = 0.613$. These results provide an interesting interpretation (and hypothesis) that inducing a ruminative cognitive state would increase the prediction performance of our dmPFC-based predictive model. We believe that these new analyses give more functional interpretability to our model, yet as the reviewer points out, from this study alone, we cannot shed light on possible additional or broader roles of the dmPFC that may emerge more robustly in other contexts.

Supplementary Figure 6. Testing the model on an additional resting-state dataset. We tested our model on an additional dataset ($n = 60$, age = 23.35 ± 1.91 [mean \pm SD], 30 males, recruited from Suwon area similar to Studies 2 and 3), which has an interesting experimental design feature—we administered two resting-state scans (each run was 14 minutes long) before and after participants watched a short emotional movie (9 minutes and 38 seconds long). We conducted this additional model test to further test our model’s generalizability and also to see whether our model showed different prediction performances depending on different resting-state conditions⁸. This movie was about a mother meeting her daughter who passed away through virtual reality, and we selected this movie to enhance internally oriented cognitive states. Scan parameters and preprocessing steps were the same as in Study 2. We also administered the Korean version of the RRS. **(a)** After each resting-state run, we asked a few questions to participants about their cognitive and affective states during the run, and as the plots show, participants had significantly higher levels of self-relevant thought and alertness. Statistical significance was calculated with a paired t-test ($df = 59$). *: $p < .05$, ***: $p < .001$. **(b)** Our model showed a significant prediction of depressive rumination only with the post-movie resting-state data, $r = 0.228$, $p = 0.040$. With the pre-movie resting-state data, our model showed non-significant prediction, $r = -0.038$, $p = 0.613$.

Reviewer #2

“The manuscript by Kim et al, titled “A DYNAMIC FUNCTIONAL CONNECTIVITY MARKER FOR RUMINATION” and submitted to the journal Nature Communications, investigates a dynamic functional connectivity predictive model of rumination, a common symptom of depression. The study combines four small datasets to form an unique approach to develop a predictive brain marker. Overall, the study has several strengths, is well written, and shows promising results. The largest weakness in my opinion is that at points the paper feels like it is written in a vacuum, without providing justifications for the choices that they make and how those choices affect downstream interpretation of the results. I have signed this review for transparency and am happy to discuss these comments if they are unclear.”

Response: We appreciate the reviewer’s thoughtful comments, many of which concerned details of our methodological approach. In the revision, we have clarified the rationale for our choices, and we provide detailed responses regarding the reviewer’s concerns below.

R2-1. *“Does the dynamic connectivity predict better than static connectivity? I think the authors need to show results based on static connectivity. Otherwise, it is hard to justify starting with dynamic functional connectivity. Alternatively, a strong motivation for dynamic functional connectivity could be presented in the introduction. Have predictive models of static connectivity notably failed in predicting depression or rumination symptoms? The literature is mixed in terms of dynamic functional connectivity being better than static connectivity for predictive modeling and brain behavior associations in general.”*

Response: We appreciate the reviewer’s comments about the importance of providing comparisons with static connectivity. Our focus on dynamic connectivity was theoretically motivated based on the importance of dynamic features inherent to rumination. In other words, we asked whether the persistent nature of rumination (i.e., a lack of cognitive temporal variability) would be captured by the variance of the DMN dynamics (i.e., temporal persistency of the neural connectivity). Nevertheless, we agree with the reviewer that comparing the dynamic connectivity results to the static connectivity ones should be informative (as also mentioned in **R3-1**). Particularly, this can also provide a connection to previous literature that showed altered static connectivity of the dMPFC in MDD patients^{9,10}. Therefore, we added additional static connectivity results to the manuscript in **Supplementary Table 2**. The results show that the models based on static connectivity did not show significant prediction performances across datasets. Therefore, across multiple datasets, the dynamic connectivity of the dMPFC showed better prediction performances for trait rumination than static connectivity. We have modified the manuscript to both reflect these new findings and clarify our *a priori* decision of using dynamic connectivity as input features.

Revision to the manuscript, p.4 (Introduction):

In light of this important feature of rumination, we hypothesized that the variance of dynamic functional connectivity would serve as an important predictor of rumination. Static or averaged dynamic DMN connectivities would reflect whether the connections between regions are high or low during the resting-state scan and have been one of the key

functional brain features for characterizing multiple clinical conditions ^{9,11,12}. However, such static connectivity measures cannot capture how stable or variable they are over time, which we hypothesized to be a key feature of rumination. Despite the importance of the variance of the connectivity, only a few studies have explored the relevance of such measures to rumination ^{13,14}. In addition, no studies of which we are aware have developed dynamic connectivity-based predictive models that can predict trait rumination in new participants.

Revision to the manuscript, p.7 (Results):

*We also trained and tested the models with static connectivity as input features, but none of the models survived (**Supplementary Table 2**), suggesting that the dynamic functional connectivity is more sensitive to individual differences in rumination than static connectivity.*

Supplementary Table 2. Training and testing results using static connectivity.

Seeds	Training (n = 84)						Test1 (n = 61)			
	Brood		Depressive		Reflective		Brood		Depressive	
	corr	permP	corr	permP	corr	permP	corr	permP	corr	permP
dMPFC	0.087	0.214	0.131	0.116	0.221	0.022				
vmPFC	0.018	0.440	0.129	0.122	-0.344	0.999				
HF (L)	0.037	0.371	0.101	0.188	0.024	0.414				
HF (R)	-0.032	0.615	-0.063	0.723	-0.073	0.748				
LTC (L)	0.158	0.074	0.040	0.360	-0.248	0.989				
PCC (L)	0.107	0.167	-0.185	0.954	-0.193	0.959				
PHC (L)	0.079	0.234	-0.130	0.883	-0.006	0.518				
Rsp (L)	0.142	0.098	-0.065	0.716	-0.175	0.942				
TPJ (L)	0.052	0.326	-0.113	0.847	-0.200	0.967				
LTC (R)	0.244	0.017	-0.175	0.944	-0.109	0.838				
PCC (R)	0.284	0.005	-0.034	0.623	0.045	0.346				
PHC (R)	-0.085	0.778	-0.187	0.955	-0.287	0.996				
Rsp (R)	-0.268	0.992	-0.277	0.995	-0.351	1.000				
TPJ (R)	-0.122	0.862	-0.031	0.614	-0.082	0.769				
pIPL (L)	0.418	0.000	0.377	0.000	0.262	0.010	-0.039	0.615	0.024	0.432
TempP (L)	0.219	0.019	-0.088	0.786	0.234	0.017				
aMPFC (L)	-0.071	0.737	-0.119	0.859	0.054	0.309				
pIPL (R)	-0.093	0.796	-0.116	0.854	-0.090	0.788				
TempP (R)	0.234	0.014	0.151	0.088	0.281	0.006				
aMPFC (R)	0.192	0.041	0.176	0.053	0.191	0.038				

Note. Using the 20 regions-of-interest (ROIs) within the default mode network, we trained and tested predictive models using static functional connectivity as input features. We corrected the significance for the multiple tests with the false discovery rate (FDR) $q < .05$ ($p < 1e-4$). (L): Left; (R): Right. dmPFC: Dorsomedial prefrontal cortex, vmPFC: Ventromedial prefrontal cortex, HF: Hippocampal formation, LTC: Lateral temporal cortex, PCC: Posterior cingulate cortex, PHC: Parahippocampal cortex. Rsp: Retrosplenial cortex, TPJ:

Temporoparietal junction, pIPL: posterior inferior parietal lobule, TempP: Temporal pole, aMPFC: anterior medial prefrontal cortex.

R2-2: *“Similarly, the choice of Dynamic Conditional Correlation (DCC) should be explained. How is this better suited for the task at hand over the large number of dynamic functional connectivity methods currently available.”*

Response: Our choice of using the dynamic conditional correlation (DCC) was motivated by Lindquist et al., 2014¹⁵, which conducted extensive tests with multiple dynamic connectivity methods including 1) a sliding-time window method with various window sizes, 2) a tapered sliding-window method, 3) an exponential weighted moving average method, and 4) the DCC. The results showed that the DCC was the best in detecting dynamic variability, which was our main target as well. In addition, the DCC has a strength that it doesn't require arbitrary choice of hyperparameters (e.g., size of the sliding windows). In our revision, we added the justification of our choice of the DCC.

Revision to the manuscript, p.25 (Methods):

We chose to use the DCC based on results from Lindquist et al. ¹⁵, which tested multiple dynamic functional connectivity methods including the sliding-time windows method with various window sizes and reported that the DCC showed a good balance between the sensitivity and specificity in estimating the variance of dynamic functional connectivity. In addition, the DCC is known to have a higher level of test-retest reliability in estimating the variance¹⁶ and does not require any additional arbitrary hyperparameters.

R2-3: *“While targeting the default mode networks is well justified, why are all seed regions treated independently rather than combining or mix & matching different seeds? Combine all the imaging features together into a single model may result in a more generalizable model.”*

Response: Our seed-based approach stems from our efforts to exploit previous literature. Previous literature on rumination has shown results ranging from the network level, sub-network level, and regional level. We reasoned that the mixed findings at the network and sub-network level studies require a regional level investigation. In particular, the dmPFC has appeared to be a key DMN region for rumination in previous literature, and we wanted to contribute to the region-level investigation by testing the dmPFC in the predictive modeling context. We agree that combining all the features would provide a better performance, but we believe that our seed-based approach can ensure better interpretability. The next comment is also relevant. Therefore, please see our continued response below.

R2-4: *“Relatedly, the authors appear to prefer sparsity in their approach. The lasso model, the validation approach of choosing the “winning” models at each dataset, the focusing on a single seed of the DMN at a time, and the refinement step will all lead to a small number of features being retained in a single model. Which are all fine, but these choices should be discussed. In particular, there is a*

confirmation bias of sorts in the discussion. The authors chose to look at the DMN because of its role in depression, find that a key node of the DMN is predictive of depression, and then discuss the DMN role in depression. But a different approach that predict similarly well with different brain regions would lead to a different discussion. I guess is this all just to say that I feel that the authors should caution how their modeling choices lead to the studies neurobiological features.”

Response: We thank the reviewer for this comment, which gave us a chance to justify our approach and clarify the implication of our findings in more depth. A big advantage prioritized in the current study is the ability to build on previous concepts and integrate the findings with previous literature (including animal literature), the majority of which are region- or pathway-centered. We appreciate the tension with more integrative models with potentially greater predictive power, but region-based models with sparse connections allow us to directly integrate with prior literature in a way that might otherwise be impossible. We are sensitive to the trend in the literature of selecting and confirming prior findings (i.e., selection and confirmation biases), which we have pointed out in many of our previous papers. For example, we suspect this has happened with the amygdala and threat learning, for example, and more strongly predictive regions have been missed (e.g., ref. ¹⁷). The other side of this coin is, however, is that the hypotheses-driven approach can dramatically increase power and reduce false positives. Here we attempted to strike the balance by testing the dmPFC with other regions across the DMN—basically placing the dmPFC in the distribution. In addition, given our limited power due to small sample sizes, we reasoned the current approach provides a good balance between type I and type II errors. In our revision, we added more detailed discussion on this issue.

Revision to the manuscript, p. 4 (Introduction):

Interpreted in this framework, a dorsal medial system is the most reasonable candidate important for the repetitive high-level appraisal that characterizes ruminative thinking. Particularly, the dorsal medial prefrontal cortex (dmPFC) has been suggested to be a core brain region for rumination. It has been considered a “dorsal nexus” that serves as a core that modulates the connectivity related to depression¹¹, and the heightened connectivity of the dmPFC was a unique feature of major depressive disorder (MDD) compared to other mental disorders¹² (see also refs. ^{18,19}). We, therefore, hypothesized that dmPFC connectivity may be a key brain region important for trait rumination. However, there have also been some inconsistent reports in the literature. For example, the static functional connectivity strength within the dorsal medial system was increased⁹ or decreased²⁰ in individuals with MDD (see also ref.^{21,22}). Collectively, these findings highlight the need for adopting a predictive modeling approach, which could provide more reliable results ²³. Furthermore, in addition to the dmPFC, we also tested other brain regions across the DMN to minimize the potential bias in our findings.

Revision to the manuscript, p.18-19 (Discussion):

First, the current study aimed to balance sparsity with complexity by focusing on multiple seed regions across the DMN, by permitting target regions across the entire brain, and by employing Lasso regression to build predictive models of rumination. Although restricting the search space to DMN seeds may increase selection and confirmation bias in our findings, the approach builds on a rich human and animal literature pointing to the role of the DMN in processes relating to emotion and rumination, and allows for easier interpretation when

integrating our findings with prior literature in a way that might otherwise be impossible with models allowing for increased complexity. Our approach, which allows for both hypothesis-driven and exploratory findings, is also advantageous in its ability to increase power, reduce false positives, and balance type I and type II errors. However, future studies, especially if well-powered, could benefit from a more exploratory approach (e.g., combining multiple models, including more features, using less sparse models, etc.).

R2-5: *“Why were each dataset chosen for their specific role (training, testing, validation)? While the training dataset has the largest sample size, the sample sizes of all datasets are of the same order. If the authors iterated through every combination of assigning a dataset to training, testing, and validation, would the same models “win”? Along the same lines, why did the authors take this approach to cross-validation compared to a more standard 10-fold cross validation (even if in the supplement) could be a value to the field.”*

Response: We determined each dataset’s role *a priori*, and the reasons were as follows. The first and most important reason was the chronological order of the data collection. That is, we collected the training dataset (Study 1) first, and then the validation and testing datasets (Studies 2 and 3). We started this project when we had only the Study 1 dataset, and thus the decision of using the dataset for training was natural. Second, as the reviewer noted, the Study 1 dataset has the largest sample size ($n = 84$, compared to $n = 61$ and 48 for Studies 2 and 3, respectively) and is also ethnically most diverse, i.e., 34 Non-Hispanic White Americans, 30 Hispanic White Americans, and 20 African American. Also, Study 1 data had the largest variance in depressive rumination scores (**Supplementary Fig.3**). Given that increasing training data variability usually improves the generalizability of models, using Study 1 data as the training dataset is reasonable. We made the reasons for our *a priori* choices clearer in our revision.

Revision to the manuscript, p. 26 (Methods):

*Our decision to use Study 1 as the training dataset was made a priori because it had the largest sample size ($n = 84$), the most racial-ethnic diversity (see ‘Participants’ section), and the largest variance in trait rumination scores across participants (**Supplementary Fig. 3**), all of which can help improve the generalizability of our model.*

Supplementary Figure 3. Distributions of the depressive rumination scores across datasets. Study 1 dataset showed the largest variance in the depressive rumination scores. For Study 1, mean = 16.69, SD = 5.627; for Study 2, mean = 13.03, SD = 4.324; for Study 3, mean = 13.77, SD 4.323..

In addition, following the reviewer’s suggestion, we conducted additional analyses to examine whether shuffling the role of datasets would yield similar results. We added the results to the manuscript as **Supplementary Tables 3-4** (see below), which showed that no models survived the generalization test. We also tested 10-fold cross-validation, which also yielded non-significant results ($r = .106$, $p = .164$, permutation test). These results seemed discouraging, but we reasoned that it could be due to the methodological characteristics of Lasso regression. For example, to minimize overfitting, we did not conduct hyperparameter tuning (i.e., lambda) in Lasso regression, which means that we used the Lasso regularization only to reduce the number of predictors to $n_{\text{obs}} - 1$, which is the largest number of predictors that we can keep. Because of this, the number of predictors became dependent on the number of observations (i.e., sample size) of the training dataset. Then, if the successful prediction of rumination requires at least a certain number of features, the shuffling and 10-fold cross-validation would fail because, compared to the model trained on Study 1 dataset, which has the largest sample size, the models trained on Study 2 and Study 3 datasets will have a smaller number of predictors. Similarly, a 10-fold cross-validation that used 90% of the sample for training in each fold ended up testing models with a fewer number of predictors compared to the leave-one-subject-out cross-validation.

To test this hypothesis, we combined Study 2 and 3 datasets ($n = 109$) and trained a model using the combined dataset, and used Study 1 dataset ($n = 84$) for independent testing. In addition, we tested the models with varying numbers of predictors and also tested 10-fold cross-validation. The combined dataset provides a sample size large enough to keep the comparable number of predictors as the original training dataset even with the 10-fold cross-validation, allowing us to test whether the successful prediction of rumination indeed requires at least a certain number of predictors to be retained. As shown in a figure below (which we added to the manuscript as **Supplementary Fig. 4**),

we found that the dmPFC-based predictive model showed significant prediction performance in both training and testing datasets only when the number of predictors exceeded 80 (**Supplementary Fig. 4**). This can explain why training models with a small sample-size datasets (i.e., Studies 2 and 3, and 10-fold cross-validation) failed to show significant model performance. We added these results and discussions to our revision.

Revision to the manuscript, p. 7-9 (Results):

*In addition, to examine the robustness of our results, we repeated our analysis by shuffling and combining the training, validation, and testing datasets. When we trained predictive models with the Study 2 or Study 3 dataset alone, we could not replicate the results (**Supplementary Tables 3 and 4**). However, when we trained the models with the combined dataset of Studies 2 and 3 ($n = 109$) and tested the model on the Study 1 dataset, we were able to replicate the original results—the dmPFC-based predictive model showed significant prediction performance in both training and testing datasets, but it did so only when the number of predictors was greater than 80 (**Supplementary Fig. 4**). These additional analyses suggest that the successful prediction of rumination requires at least a certain number of predictors.*

Supplementary Table 3. Prediction results using Study 2 as a training dataset.

Seeds	Training ($n = 61$)						Test1 ($n = 84$)					
	Brood		Depressive		Reflective		Brood		Depressive		Reflective	
	corr	permP	corr	permP	corr	permP	corr	permP	corr	permP	corr	permP
dmPFC	0.191	0.072	-0.142	0.862	0.396	0.001					0.042	0.351
vmPFC	0.398	0.001	-0.173	0.907	0.007	0.467	0.066	0.273				
HF (L)	0.091	0.241	0.025	0.422	0.429	0.001					-0.216	0.978
HF (R)	-0.201	0.941	-0.304	0.991	-0.209	0.954						
LTC (L)	0.209	0.053	0.048	0.349	-0.211	0.946						
PCC (L)	-0.039	0.625	-0.098	0.775	-0.139	0.848						
PHC (L)	-0.198	0.937	-0.227	0.964	0.366	0.001					-0.076	0.762
Rsp (L)	0.274	0.015	0.040	0.383	-0.155	0.884						
TPJ (L)	0.193	0.068	0.103	0.216	-0.079	0.720						
LTC (R)	-0.276	0.984	0.171	0.097	-0.100	0.787						
PCC (R)	-0.204	0.943	0.175	0.089	-0.223	0.959						
PHC (R)	0.060	0.339	0.302	0.009	0.258	0.022						
Rsp (R)	0.154	0.113	0.009	0.470	0.384	0.001					-0.120	0.861
TPJ (R)	0.237	0.032	0.308	0.009	-0.231	0.966						
pIPL (L)	-0.059	0.673	0.157	0.113	0.349	0.004					-0.170	0.938
TempP (L)	0.172	0.090	0.323	0.004	0.442	0.000			-0.002	0.511	0.078	0.238
aMPFC (L)	-0.076	0.713	0.334	0.004	0.226	0.038			-0.036	0.617		
pIPL (R)	-0.101	0.783	-0.002	0.518	-0.057	0.661						
TempP (R)	0.208	0.054	-0.205	0.936	0.201	0.059						
aMPFC (R)	0.080	0.273	0.262	0.021	-0.006	0.519						

Note. Using the same 20 default mode network regions-of-interest as in Table 1, we trained and tested the models using the variance of seed-based dynamic functional connectivity. The difference is that this shows the results of using Study 2 as a training dataset. We corrected for multiple comparisons with the false discovery rate ($p < .009$ for FDR $q < .05$). Here, we used Study 1 dataset for validation. (L): Left; (R): Right. dmPFC: Dorsomedial prefrontal cortex, vmPFC: Ventromedial prefrontal cortex, HF: Hippocampal formation, LTC: Lateral temporal cortex, PCC: Posterior cingulate cortex, PHC: Parahippocampal cortex. Rsp: Retrosplenial cortex, TPJ: Temporoparietal junction, pIPL: posterior inferior parietal lobule, TempP: Temporal pole, amPFC: anterior medial prefrontal cortex

Supplementary Table 4. Prediction results using Study 3 as a training dataset.

Seeds	Training ($n = 48$)						Test1 ($n = 84$)				Test2 ($n = 61$)	
	Brood		Depressive		Reflective		Brood		Reflective		Brood	
	corr	permP	corr	permP	corr	permP	corr	permP	corr	permP	corr	permP
dMPFC	0.009	0.473	-0.056	0.656	0.158	0.144						
vMPFC	0.411	0.002	0.215	0.073	-0.175	0.880						
HF (L)	-0.108	0.771	-0.004	0.513	0.110	0.224						
HF (R)	0.527	0.000	0.152	0.155	-0.083	0.721	0.187	0.041			-0.067	0.692
LTC (L)	-0.150	0.845	-0.195	0.911	0.292	0.022						
PCC (L)	-0.478	1.000	0.101	0.250	0.390	0.004						
PHC (L)	0.289	0.023	0.145	0.169	0.436	0.002			0.075	0.249		
Rsp (L)	-0.306	0.983	-0.194	0.903	-0.323	0.989						
TPJ (L)	0.025	0.432	0.082	0.297	-0.414	0.999						
LTC (R)	-0.143	0.835	-0.280	0.971	0.077	0.297						
PCC (R)	0.351	0.007	-0.285	0.975	-0.221	0.934						
PHC (R)	0.064	0.332	-0.290	0.980	-0.515	1.000						
Rsp (R)	-0.043	0.610	0.339	0.008	0.088	0.263						
TPJ (R)	0.026	0.422	-0.111	0.777	-0.227	0.937						
pIPL (L)	0.344	0.009	-0.136	0.816	0.264	0.037						
TempP (L)	-0.042	0.620	0.004	0.490	-0.421	0.999						
aMPFC (L)	0.160	0.139	-0.207	0.914	0.094	0.262						
pIPL (R)	-0.348	0.992	0.162	0.137	0.217	0.065						
TempP (R)	-0.285	0.973	-0.195	0.909	-0.132	0.813						
aMPFC (R)	-0.091	0.734	-0.072	0.680	-0.214	0.934						

Note. Same with Supplementary Table 3, except that this shows the results of using Study 3 as a training dataset.

Supplementary Figure 4. Prediction performance of the dmPFC-based predictive model with varying numbers of predictors. Here we trained the dmPFC-based predictive model with a combined dataset of Study 2 and 3 ($n = 109$) to examine whether the successful prediction of rumination required at least a certain number of predictors. The testing dataset was the Study 1 dataset ($n = 84$). We varied the number of predictors to be retained in the Lasso regression and plotted the correlation between the model response (y_{fit}) and the dependent variable (y). For the training dataset, the y - y_{fit} correlation was from 10-fold cross-validation. The results show that the dmPFC-based predictive model showed significant prediction performance in both training and testing datasets, but it did so only when the number of predictors was greater than 80. $*p < .05$, permutation test in both training and testing datasets.

R2-6: “Finally, why not combine all the behavioral data into a single model? As the predictive model show pretty good cross prediction of results, the brain signatures may not be independent and predicting a latent factor (say from a principal component analysis) of all measures often produces a better prediction results.”

Response: There were two main reasons that we modeled each subscale separately. First, we did so based on previous literature on rumination. Treynor et al.²⁴ showed that the reflective pondering and brooding subscales of the RRS were unconfounded with depression and differentially related to rumination. Treynor et al. thus explicitly recommended investigating the subscales separately (see the quote below).

“Specifically, it is critical to differentiate between a reflection component of rumination and a brooding component, and to measure them separately, because they have quite different relationships to depression.” (p. 258, ref. ²⁴)

Meanwhile, the depressive rumination subscale is known to include some overlapping items with the Beck Depression Inventory^{25,26}. It is still unclear whether this overlap reflects a fundamental similarity between two constructs (i.e., rumination and depression) or just a methodological confound. For these reasons, rather than lumping all these heterogeneous subscales together, we decided to model them separately.

Second, Studies 2 and 3 used the Korean version of the RRS (KRRS), which has a factor structure that is slightly different from the original RRS, which is used in Study 1. In more detail, Kim et al.²⁷ conducted an exploratory factor analysis of the KRRS and found that the KRRS also consisted of the same three subscales as the original English version of the RRS. However, the items of each subscale were slightly different as detailed in the Methods section. Combining the subscales can obscure the differences between two versions of the RRS, and thus we decided to use each subscale separately for the model development.

We addressed a similar issue in **R3-5** below, in which we described the three RRS subscales in more detail.

R2-7: *“I appreciate adding a depression dataset as a final validation. But I worry about the low number of individuals (n = 35) with depression, especially given the heterogeneous nature of depression. There are multiple depression datasets available fully processed and open sources (see REST-meta-MDD Consortium) as well as the other sites in the Strategic Research Program for Brain Sciences dataset.”*

Response: We thank the reviewer for this comment on this important issue, and we are totally with the reviewer on this concern. Of course, we also wanted to test our model on more depression datasets, but there were some reasons we decided not to do that. As the reviewer noted, all depression data are highly heterogeneous. Depression itself is a highly heterogeneous clinical condition²⁸, and also fMRI datasets from multiple scan sites should be highly heterogeneous due to their differences in scanners, sequence parameters, and many other factors²⁹. Thus, when we were planning on this final validation on a clinical dataset, we took a focused approach (again) instead of an exploratory approach because we reasoned that we do not have that much power given the high degree of variance intrinsic to the clinical fMRI data. In other words, our concern was that the exploratory tests would increase the type II error (false negative rate) dramatically. This was why we chose one clinical data with the largest sample size and from one scan site, tested our model on the dataset, and stopped. This did not mean that we did not want to test our model on other clinical datasets given that we were planning to make our rumination model freely available so that any researchers (including us) could freely test our model on their own datasets. Regardless of these concerns, however, we tested our model on more clinical datasets to follow the reviewer’s suggestion.

One thing to note is that we could not test our model on the REST-meta-MDD Consortium data because they did not provide the raw fMRI data that we need to calculate the DCC values. They only provide fully processed fMRI indices (e.g., the amplitude of low-frequency fluctuations, Regional Homogeneity, static functional connectivity, etc.) using a specific brain parcellation atlas that does not match with ours. Thus, we tested our model on three datasets from the Strategic Research Program

for Brain Sciences (SRPBS) data that had both fMRI and behavioral (i.e., BDI-II) data. These datasets were independently obtained from different scan sites and with different scan parameters.

After we applied the selection criteria same as our original analysis (i.e., participants with mean framewise displacement under 0.25 and right-handed), we were able to proceed with the following datasets: a dataset from Hiroshima Kajikawa Hospital (HKH; $n = 21$), Hiroshima University Hospital (HUH; $n = 57$), and University of Tokyo (UTO; $n = 22$). We calculated the dmPFC-based DCC variances, applied the refined model (i.e., model with 21 important regions), and compared the model prediction with the BDI-II scores. As shown in the figure below (which we have now added to our revised manuscript as **Supplementary Fig. 5**), our model did not show generalization across three datasets.

There could be many reasons for this, but according to Yamashita et al.²⁹, the measurement bias caused by different scan parameters (esp., phase encoding direction) and MRI manufacturers could be major contributors to the generalization failure. Yamashita et al.²⁹ conducted detailed analyses on the SRPBS data to minimize the heterogeneity across multiple scan sites (i.e., data harmonization) and showed that phase encoding and MRI manufacturer were the two most significant contributors to the measurement bias.

The COI (Center of Innovation in Hiroshima University) dataset that we included in the original manuscript was the only dataset that had the same phase encoding direction (i.e., A→P direction) and the same MRI manufacturer (i.e., Siemens) as ours (i.e., Studies 1-3). Unfortunately, no other depression datasets from the SRPBS used the same scan parameters and the same scanner. For example, the HUH, UTO, and HKH datasets used a different phase encoding direction (i.e., P→A), and the HUH and UTO datasets used an MRI scanner from a different manufacturer (i.e., GE). In our revision, we added these additional analyses and discussions on the limitation of our results.

Revision to the manuscript, p. 15 (Results):

*However, please note that our model fails to generalize in three other MDD patients' datasets ($n = 21, 57, \text{ and } 22$; **Supplementary Fig. 5**). Critical differences between the datasets that our model worked on versus those that did not work include phase encoding direction and MRI manufacturer, which Yamashita et al.²⁹ reported were the two most significant contributors to the measurement bias.*

Revision to the manuscript, p. 20 (Discussion):

We also showed that our model does not generalize to the clinical datasets with different scan parameters, such as phase encoding direction and MRI manufacturer. ... Also we could not directly test our model to predict rumination in MDD patients because the clinical datasets did not have the RRS scores, making it difficult to know what exactly our model predicted in depressed patients. Thus, future studies should investigate the boundary conditions when our model does work or does not work.

Supplementary Figure 5. Prediction results in additional clinical samples. To examine whether our model generalizes to other clinical depression datasets, we tested our model on three datasets from the Strategic Research Program for Brain Sciences (SRPBS) data that had both fMRI and behavioral (i.e., BDI-II) data. After we applied the selection criteria same as our original analysis (i.e., participants with mean framewise displacement under 0.25 and right-handed), we were able to proceed with the following datasets: a dataset from Hiroshima Kajikawa Hospital (HKH; $n = 21$), Hiroshima University Hospital (HUH; $n = 57$), and University of Tokyo (UTO; $n = 22$). We calculated the dmPFC-based DCC variances, applied the refined model (i.e., model with 21 important regions), and compared the model prediction with the BDI-II scores. The results showed that our model failed to generalize in other depression datasets. There could be many reasons for this, but according to Yamashita et al.²⁹, the measurement bias caused by different scan parameters (esp., phase encoding direction) and MRI manufacturers could be major contributors to the generalization failure. Yamashita et al.²⁹ conducted detailed analyses on the SRPBS data to minimize the heterogeneity across multiple scan sites (i.e., data harmonization) and showed that phase encoding and MRI manufacturer were the two most significant contributors to the measurement bias. The COI (Center of Innovation in Hiroshima University) dataset that we included in the main manuscript was the only dataset that had the same phase encoding direction (i.e., A→P direction) and same MRI manufacturer (i.e., Siemens) as ours (i.e., Studies 1-3 and Supplementary dataset). Unfortunately, no other depression datasets from the SRPBS did not use the same scan parameters and same scanner. For example, the HUH, UTO, and HKH datasets used a different phase encoding direction (i.e., P→A), and the HUH and UTO datasets used MRI scanner from a different manufacturer (i.e., GE). In our revision, we added these additional analyses and discussions on the limitation of our results.

R2-8: *There appears to be an outlier in independent test sample 2. It likely doesn't affect the results, but the authors may want to check."*

Response: Thank you for this. Removing the outlier in Study 3 (which is the independent test sample 2) did not significantly affect our results, $r = 0.276, p = 0.028$. We modified the figure, the corresponding caption, and the manuscript to make it apparent.

Revision to the manuscript, p. 6 (Figure 1.)

Revision to the manuscript, p. 7 (Caption of Figure 1.):

A red-dashed circle indicates the data point that was identified as an outlier (i.e., greater than 3 standard deviations away from the mean), which did not affect the model significance after its removal.

Revision to the manuscript, p. 7 (Results):

When we removed one outlier in the independent test dataset, the dmPFC-based model still showed significant prediction performance, $r = 0.276$, $p = 0.028$.

R2-9: “A more minor point: I think the paper could increase the diversity of citations as opposed to works from the author’s extended group. For example, it could be beneficial to cite: *Taxali et al 2021* when discussing that models have greater reliability of than edge level features.”

Response: We are grateful to the reviewer for suggesting citing an interesting paper, which has shown higher reliability of multivariate predictive modeling than edge-level features themselves. We think the paper could highlight the importance of our findings and the predictive modeling approach.

Revision to the manuscript, p. 4 (Introduction):

However, there have also been some inconsistent reports. For example, the static functional connectivity strength within the dorsal medial system was increased⁹ or decreased²⁰ in individuals with MDD (see also ref.^{21,22}), highlighting the need for adopting a predictive modeling approach, which could provide more reliable results²³.

Reviewer #3

R3-1: *“From this reviewer’s perspective, the best aspect of the manuscript is the pattern of dmPFC findings. Across cultures and in the context of assessing the nebulous and ephemeral phenomenon of rumination, it does seem like something about the dmPFC is important. The potential importance of this region as opposed to other DMN structures is presented well in the introduction. In developing the work, the authors might choose to determine if other dmPFC connectivity metrics do or do not predict depressive rumination. If they do, then we might conclude that dmPFC relations with other structures are important in general; if not, then we might reach the even more useful conclusion that connective variability of dmPFC is especially important.”*

Response: We thank the reviewer for pointing out the significance of the dmPFC findings, and we agree! We also agree that testing other connectivity metrics is important to provide a more definitive interpretation of the results. In our revised manuscript, we have added the results of using static connectivity as input features for the prediction of RRS scores (**Supplementary Table 2.**). The results show that the models based on static connectivity did not show significant prediction performances across datasets, suggesting that using the variance of dynamic connectivity is a key to the successful prediction of individual differences in rumination. Since the comment is closely related to **R2-1** above, please find our revisions in **R2-1**.

R3-2: *“Less good are the biomarker claims made and implied throughout the manuscript. The results are probably consistent but they are not particularly strong, with many parameters coming together to explain 10% of the variance. Indeed, up until the model predicts BDI in the depressed sample, the work smacks of type-I error with seven of 60 models surviving round 1 and one of seven surviving round 2 all the while using lenient FDR corrections and one-sided statistical tests.”*

Response: We appreciate the reviewer’s comment. We agree that the small effect sizes and low statistical power are our main weaknesses. We made multiple efforts to address this comment. First, we revised the title and the manuscript to tone down our biomarker claim, and instead focused more on the dmPFC findings. For example, we changed the title of the manuscript as the following:

Revision to the title:

“A dorsomedial prefrontal cortex-based dynamic functional connectivity model of rumination”

and we also provided a more in-depth discussion on the dmPFC in the Introduction:

Revision to the manuscript, p. 4 (Introduction):

“Interpreted in this framework, a dorsal medial system is the most reasonable candidate important for the repetitive high-level appraisal that characterizes ruminative thinking. Particularly, the dorsal medial prefrontal cortex (dmPFC) has been suggested to be a core brain region for rumination. It has been considered a “dorsal nexus” that serves as a core that modulates the connectivity related to depression¹¹, and the heightened connectivity of the dmPFC was a unique feature of major depressive disorder (MDD) compared to other mental disorders¹² (see also refs. ^{18,19}). We, therefore, hypothesized that the dmPFC

connectivity may be a key brain region important for trait rumination. However, there have also been some inconsistent reports. For example, the static functional connectivity strength within the dorsal medial system was increased⁹ or decreased²⁰ in individuals with MDD (see also ref.^{21,22}), highlighting the need for adopting a predictive modeling approach, which could provide more reliable results²³. Furthermore, in addition to the dmPFC, we also tested other brain regions across the DMN to minimize the potential bias in our findings.”

In addition, we have added one more dataset as independent test data to further test the generalizability of our model. This dataset that we recently finished collecting (during the revision period) has an interesting experimental design feature, which is that we administered two 14-min resting-state runs before and after participants watched a short emotional movie (around 10 mins long). This movie was about a mother meeting her daughter who passed away through virtual reality, and we selected this movie to enhance internally oriented cognitive states. As shown in the figure below (which is now added to the manuscript as **Supplementary Figure 6**), our model showed a significant prediction of depressive rumination only with the post-movie resting-state data, $r = 0.228$, $p = 0.040$. With the pre-movie resting-state data, our model showed non-significant prediction, $r = -0.038$, $p = 0.613$. Again, the results showed a small effect size and one negative result, which could still suggest a possibility of type-I error. However, this also provides an interesting hypothesis that inducing a ruminative cognitive state would increase the prediction performance of our model, which should be examined in future studies. Overall, though we cannot provide definitive evidence that our findings are not false positives, we at least put our best effort into further testing our model and here provide one positive and one negative result with an interesting hypothesis for future study.

Revision to the manuscript, p. 19 (Discussion):

*our prediction results showed only small effect sizes, and our sample sizes were small for predicting individual differences³⁰, suggesting a possibility of type I errors in this study. To further test the generalizability of our model, we tested our model on an additional subclinical independent dataset ($n = 60$; see captions of **Supplementary Fig. 6** for the details of this dataset) that we finished collecting during revision. This Supplementary dataset had an interesting experimental design feature—we administered two 14-min resting-state runs before and after participants watched a short emotional movie (around 10 mins long). This movie was about a mother meeting her daughter who passed away through virtual reality, and we selected this movie to enhance internally oriented cognitive states. As shown in **Supplementary Figure 6**, our model showed a significant prediction of depressive rumination only with the post-movie resting-state data, $r = 0.228$, $p = 0.040$. With the pre-movie resting-state data, our model showed a non-significant prediction, $r = -0.038$, $p = 0.613$. Again, the results showed a small effect size and one negative result, which could still suggest a possibility of type-I error. However, this also provides an interesting hypothesis that inducing a ruminative cognitive state would increase the prediction performance of our model, which should be examined in future studies. Overall, though we cannot provide definitive evidence that our findings are not false positives in the current study, we put our*

best effort into further testing our model, and here we provide one positive and one negative result with an interesting hypothesis for future study.

Supplementary Figure 6. Testing the model on an additional resting-state dataset. We tested our model on an additional dataset ($n = 60$, age = 23.35 ± 1.91 [mean \pm SD], 30 males, recruited from Suwon area similar to Studies 2 and 3), which has an interesting experimental design feature—we administered two resting-state scans (each run was 14 minutes long) before and after participants watched a short emotional movie (9 minutes and 38 seconds long). We conducted this additional model test to further test our model’s generalizability and also to see whether our model showed different prediction performances depending on different resting-state conditions⁸. This movie was about a mother meeting her daughter who passed away through virtual reality, and we selected this movie to enhance internally oriented cognitive states. Scan parameters and preprocessing steps were the same as in Study 2. We also administered the Korean version of the RRS. **(a)** After each resting-state run, we asked a few questions to participants about their cognitive and affective states during the run, and as the plots show, participants had significantly higher levels of self-relevant thought and alertness. Statistical significance was calculated with a paired t-test ($df = 59$). *: $p < .05$, ***: $p < .001$. **(b)** Our model showed a significant prediction of depressive rumination only with the post-movie resting-state data, $r = 0.228$, $p = 0.040$. With the pre-movie resting-state data, our model showed non-significant prediction, $r = -0.038$, $p = 0.613$.

R3-3: “The authors state, “We therefore hypothesize that components of the dorsal medial system—especially connectivity with the dmPFC—may be a key neural marker important for both trait and state rumination.” How might a region of the brain mark both a cognitive phenomenon in action as well as the tendency for that phenomenon to occur in the absence of that phenomenon? This is important to consider in light of claims that the imaging results might provide “a direct window into depression-relevant brain processes.” This is especially noteworthy in consideration of the lack of reported results from the thought-probe resting fMRI approach from Study 3. This scan would seem to provide the only

means close to assessing rumination at the state level.”

Response: We thank the reviewer for the comment. We think that it is difficult to specify the functional role of the dmPFC (see **R1-4** as well) with our data, and thus we deleted the sentence. Though we still believe that the dmPFC could play a crucial role in both state and trait rumination based on previous literature^{31,32}, we decided to narrow our focus on modeling trait rumination given that our outcome measure, the RRS, targets the trait rumination. However, we were also intrigued by the reviewer’s question, “*How might a region of the brain mark both a cognitive phenomenon in action as well as the tendency for that phenomenon to occur in the absence of that phenomenon*”. We reasoned that we could approach this question in two ways. First, we could develop an additional predictive model based on Study 3. Second, we could examine whether our model also capture state rumination using thought-sampling task data from Study 3. As we decided to focus on trait rumination, we did not choose the first approach. In addition, the second approach had two challenges: First, each thought-sampling trial in Study 3 lasted only for a short duration (45 seconds), which could result in unreliable estimates of functional connectivity. Second, we had no in-scanner report for rumination to directly compare the model response with, making it challenging to know whether the model is predicting the construct of interest or not. Due to these limitations, we instead decided to test our model on an additional supplementary dataset as described in **R3-2** above.

R3-4: *“In one sense, the authors’ model predicting BDI in a diagnosed depressed sample is quite nice in that the model was trained on rumination of three varieties, only one of which associates strongly with levels of depressive symptoms which, themselves, fluctuate daily in depressed persons. In another sense, it’s a stretch to contend that dmPFC DCC with several brain regions is a marker of rumination in depression when rumination in depression was not formally assessed.”*

Response: We agree with the reviewer and thus we added the following sentence in the discussion.

Revision to the manuscript, p. 21 (Discussion):

Also, we could not directly test our model to predict rumination in MDD patients because the clinical datasets did not have the RRS scores, making it difficult to know what exactly our model predicted in depressed patients.

R3-5: *“To be faithful to the constructs of rumination presented, the authors should make clear that what they are calling “depressive rumination” is rumination that is correlated to levels of depressive symptoms in a depressed cohort studied by Treynor et al. The original notion was to study rumination in depression that was not part and parcel of depressive symptomatology—i.e., brooding and pondering.”*

Response: We appreciate the reviewer’s comment about the need to clarify what depressive rumination is. We admit that the term ‘depressive rumination’ has caused some confusion. We changed the citation when explaining RRS, and further added some clarification comments in the Methods section as below:

Revision to the manuscript, p.22 (Methods):

There have been some debates on which factors comprise a reliable construct of rumination³³⁻³⁵. Here, we used the sum of each subscale of RRS suggested in Treynor et al.²⁴ as a measure of each subscale construct. Treynor et al. argued that the depressive rumination subscale might be confounded with depression itself, while the other two are unconfounded with depression and each reflects adaptive (reflective pondering) and maladaptive (brooding) aspects of rumination.

R3-6: *“In Study 3, the authors “regressed out thought sampling-related fMRI signal.” This might give prospective readers the impression that the effects of conducting thought sampling five times during a putative resting-state scan did not otherwise interfere with cognitive and affective processes that occurred during the scan. Thought sampling is a necessary and potentially powerful tool but there is little to do about resulting “Heisenberg effects.” In this vein, it is possible that prompting subjects about their thought content caused more internal self-focus than would have otherwise been the case which then biased patterns of resting-brain connectivity toward the authors’ theoretical stance.”*

Response: We thank the reviewer for this insightful comment. We totally agree about the possibility of ‘prompting subjects about their thought content caused more internal self-focus’ and decided to make it as explicit as possible as seen in **R3-2**. However, a fundamental challenge here is that we have no in-scanner report of participants’ internal cognitive states (e.g., rumination), making it difficult to prove that participants were indeed in a more self-focused state or conclude that the internally oriented cognitive state leads to significant prediction performance. This should be examined in future studies.

R3-7: *“The concept of reliability is a cornerstone of the authors’ study. It is noteworthy, then, that three of the four sub-studies collected less BOLD data than the minimum recommended in order to achieve adequate test-retest reliability of rsfMRI-derived metrics (e.g., <https://www.sciencedirect.com/science/article/pii/S0149763414001262>).”*

Response: We thank the reviewer for pointing out the potential reliability issue. We agree that reliability is greatly important, and we already had a paragraph on the potential reliability issue in the discussion section. We have added the citation that the reviewer provided. However, what we have shown in our previous papers^{17,36} is that a good predictive model improves both effect sizes and reliability of model scores dramatically³⁷. Thus, achieving good reliability of individual features will always be difficult, but it is possible to get an overall pattern that is reliable.

R3-8: *“Calculating correlations between changes in delta corr across two studies in the virtual lesion analysis approach is a clever way of showing the statistical reliability of the overall pattern of “lesion” effects. In the end, though, it also shows that there is only ~40% correspondence between models.”*

Response: We appreciate your concern about the consistency of virtual lesion results. Yes, 40% correspondence seems small, but we showed that the refined model worked in the MDD patient group, suggesting that the overlaps between the two datasets were meaningful. In addition, the high delta correlation values across two datasets were found in the left IFG, suggesting the importance of the region. This finding is plausible considering previous literature^{6,38}. Thus, the virtual lesion analysis results seem to provide meaningful results.

References

- 1 Karapanagiotidis, T. *et al.* The psychological correlates of distinct neural states occurring during wakeful rest. *Sci Rep* **10**, 21121, doi:10.1038/s41598-020-77336-z (2020).
- 2 Goodman, Z. T. *et al.* Whole-Brain Functional Dynamics Track Depressive Symptom Severity. *Cereb Cortex*, doi:10.1093/cercor/bhab047 (2021).
- 3 Smallwood, J. Distinguishing How From Why the Mind Wanders: A Process-Occurrence Framework for Self-Generated Mental Activity. *Psychological Bulletin* **139**, 519-535, doi:10.1037/a0030010 (2013).
- 4 Murphy, C. *et al.* Modes of operation: A topographic neural gradient supporting stimulus dependent and independent cognition. *Neuroimage* **186**, 487-496, doi:10.1016/j.neuroimage.2018.11.009 (2019).
- 5 Zhang, M. *et al.* Perceptual coupling and decoupling of the default mode network during mind-wandering and reading. *Elife* **11**, doi:10.7554/eLife.74011 (2022).
- 6 Vatansever, D. *et al.* Varieties of semantic cognition revealed through simultaneous decomposition of intrinsic brain connectivity and behaviour. *Neuroimage* **158**, 1-11, doi:10.1016/j.neuroimage.2017.06.067 (2017).
- 7 Turnbull, A. *et al.* Left dorsolateral prefrontal cortex supports context-dependent prioritisation of off-task thought. *Nat Commun* **10**, 3816, doi:10.1038/s41467-019-11764-y (2019).
- 8 Finn, E. S. Is it time to put rest to rest? *Trends in Cognitive Sciences* **25**, 1021-1032, doi:10.1016/j.tics.2021.09.005 (2021).
- 9 Zhu, X., Zhu, Q., Shen, H., Liao, W. & Yuan, F. Rumination and Default Mode Network Subsystems Connectivity in First-episode, Drug-Naive Young Patients with Major Depressive Disorder. *Sci Rep* **7**, 43105, doi:10.1038/srep43105 (2017).
- 10 Yan, C. G. *et al.* Reduced default mode network functional connectivity in patients with recurrent major depressive disorder. *Proc Natl Acad Sci U S A* **116**, 9078-9083, doi:10.1073/pnas.1900390116 (2019).
- 11 Sheline, Y. I., Price, J. L., Yan, Z. & Mintun, M. A. Resting-state functional MRI in depression unmasks increased connectivity between networks via the dorsal nexus. *Proceedings of the National Academy of Sciences of the United States of America* **107**, 11020-11025, doi:10.1073/pnas.1000446107 (2010).
- 12 Doucet, G. E. *et al.* Transdiagnostic and disease-specific abnormalities in the default-mode network hubs in psychiatric disorders: A meta-analysis of resting-state functional imaging studies. *Eur Psychiatry* **63**, e57, doi:10.1192/j.eurpsy.2020.57 (2020).
- 13 Kaiser, R. H. *et al.* Dynamic Resting-State Functional Connectivity in Major Depression. *Neuropsychopharmacology* **41**, 1822-1830, doi:10.1038/npp.2015.352 (2016).
- 14 Wang, J. *et al.* Abnormal dynamic functional network connectivity in unmedicated bipolar and major depressive disorders based on the triple-network model. *Psychol Med* **50**, 465-474, doi:10.1017/S003329171900028X (2020).
- 15 Lindquist, M. A., Xu, Y., Nebel, M. B. & Caffo, B. S. Evaluating dynamic bivariate correlations in resting-state fMRI: A comparison study and a new approach. *Neuroimage* **101**, 531-546, doi:10.1016/j.neuroimage.2014.06.052 (2014).
- 16 Choe, A. S. *et al.* Comparing test-retest reliability of dynamic functional connectivity methods. *Neuroimage* **158**, 155-175, doi:10.1016/j.neuroimage.2017.07.005 (2017).
- 17 Reddan, M. C., Lindquist, M. A. & Wager, T. D. Effect Size Estimation in Neuroimaging. *JAMA Psychiatry* **74**, 207-208, doi:10.1001/jamapsychiatry.2016.3356 (2017).

- 18 Kaiser, R. H., Andrews-Hanna, J. R., Wager, T. D. & Pizzagalli, D. A. Large-Scale Network Dysfunction in Major Depressive Disorder: A Meta-analysis of Resting-State Functional Connectivity. *JAMA Psychiatry* **72**, 603-611, doi:10.1001/jamapsychiatry.2015.0071 (2015).
- 19 Sha, Z., Wager, T. D., Mechelli, A. & He, Y. Common Dysfunction of Large-Scale Neurocognitive Networks Across Psychiatric Disorders. *Biol Psychiatry* **85**, 379-388, doi:10.1016/j.biopsych.2018.11.011 (2019).
- 20 Tozzi, L. *et al.* Reduced functional connectivity of default mode network subsystems in depression: Meta-analytic evidence and relationship with trait rumination. *Neuroimage Clin* **30**, 102570, doi:10.1016/j.nicl.2021.102570 (2021).
- 21 Lois, G. & Wessa, M. Differential association of default mode network connectivity and rumination in healthy individuals and remitted MDD patients. *Soc Cogn Affect Neurosci* **11**, 1792-1801, doi:10.1093/scan/nsw085 (2016).
- 22 Jacob, Y. *et al.* Neural correlates of rumination in major depressive disorder: A brain network analysis. *NeuroImage: Clinical* **25**, 102142-102142, doi:10.1016/j.nicl.2019.102142 (2020).
- 23 Taxali, A., Angstadt, M., Rutherford, S. & Sripada, C. Boost in Test-Retest Reliability in Resting State fMRI with Predictive Modeling. *Cereb Cortex* **31**, 2822-2833, doi:10.1093/cercor/bhaa390 (2021).
- 24 Treynor, W., Gonzalez, R. & Nolen-Hoeksema, S. Rumination Reconsidered: A Psychometric Analysis. **27**, 247-259, doi:10.1023/A:1023910315561 (2003).
- 25 Conway, M., Csank, P. A., Holm, S. L. & Blake, C. K. On assessing individual differences in rumination on sadness. *J Pers Assess* **75**, 404-425, doi:10.1207/S15327752JPA7503_04 (2000).
- 26 Segerstrom, S. C., Tsao, J. C. I., Alden, L. E. & Craske, M. G. Worry and rumination: Repetitive thought as a concomitant and predictor of negative mood. *Cognitive Ther Res* **24**, 671-688, doi:Doi 10.1023/A:1005587311498 (2000).
- 27 Kim, S.-J., Kim, J. & Youn, S.-C. Validation of the Korean-Ruminative Response Scale(K-RRS). *Korean Journal of Clinical Psychology* **29**, 1-19, doi:10.15842/kjcp.2010.29.1.001 (2010).
- 28 Fried, E. I. & Nesse, R. M. Depression sum-scores don't add up: why analyzing specific depression symptoms is essential. *Bmc Medicine* **13**, doi:ARTN 72 10.1186/s12916-015-0325-4 (2015).
- 29 Yamashita, A. *et al.* Harmonization of resting-state functional MRI data across multiple imaging sites via the separation of site differences into sampling bias and measurement bias. *Plos Biology* **17**, doi:10.1371/journal.pbio.3000042 (2019).
- 30 Marek, S. *et al.* Reproducible brain-wide association studies require thousands of individuals. *Nature* **603**, 654-660, doi:10.1038/s41586-022-04492-9 (2022).
- 31 Chen, X. *et al.* The subsystem mechanism of default mode network underlying rumination: A reproducible neuroimaging study. *Neuroimage* **221**, 117185, doi:10.1016/j.neuroimage.2020.117185 (2020).
- 32 Zhou, H. X. *et al.* Vol. 206 116287-116287 (Academic Press Inc., 2020).
- 33 Roberts, J. E., Gilboa, E. & Gotlib, I. H. Ruminative response style and vulnerability to episodes of dysphoria: Gender, neuroticism, and episode duration. *Cognitive Ther Res* **22**, 401-423, doi:Doi 10.1023/A:1018713313894 (1998).
- 34 Lam, D., Smith, N., Checkley, S., Rijdsdijk, F. & Sham, P. Effect of neuroticism, response style and information processing on depression severity in a clinically depressed sample. *Psychol Med* **33**, 469-479, doi:10.1017/s0033291702007304 (2003).
- 35 Bagby, R. M. & Parker, J. D. A. Relation of rumination and distraction with neuroticism and extraversion in a sample of patients with major depression. *Cognitive Ther Res* **25**, 91-102, doi:Doi 10.1023/A:1026430900363 (2001).
- 36 Kragel, P. A., Han, X. C., Kraynak, T. E., Gianaros, P. J. & Wager, T. D. Functional MRI Can Be Highly Reliable, but It Depends on What You Measure: A Commentary on Elliott *et al.* (2020). *Psychological Science* **32**, 622-626, doi:10.1177/0956797621989730 (2021).
- 37 Han, X. *et al.* Effect sizes and test-retest reliability of the fMRI-based neurologic pain signature. *Neuroimage* **247**, 118844, doi:10.1016/j.neuroimage.2021.118844 (2021).
- 38 Andrews-Hanna, J. R. & Grilli, M. D. Mapping the imaginative mind: Charting new paths forward. *Curr Dir Psychol Sci* **30**, 82-89, doi:10.1177/0963721420980753 (2021).

Reviewers' Comments:

Reviewer #1:

Remarks to the Author:

I think this revision deals with all of the suggestions I made about the initial submission. I believe the paper is stronger because of these changes and as I explained in my initial review I am very excited about the paper and in particular it's convergence with other studies along similar lines. I am looking forward to seeing this paper in press in due course.

Signed,

Jonathan Smallwood

Reviewer #2:

Remarks to the Author:

Overall the authors have answered my questions/comments from the previous review.

I think there is an error in Table 1. For the last column (Test2 (n=48)), the r is listed as 0.24 but it should be 0.288 per the text above.

It could be helpful to put in the SI the tables for combining dataset 2+3 for training (maybe one for the original number of features and one for >80 features). That would make it easier to compare when and how replication is achieved across different training/testing breakdowns. It might be also worth noting this as a limitation.

I really appreciate the tables. They make it easy to compare across models.

Reviewer #4:

Remarks to the Author:

In their manuscript, Kim et al. present a multivariate predictive model of trait rumination using as input dynamic functional connectivity between the dorsomedial prefrontal cortex and the rest of the brain. Notably, the performance of this model was tested in multiple independent datasets and the results generalized in many of them.

Resting state fluctuations in regions belonging to the default mode network, such as the dorsomedial prefrontal cortex, have been shown to be related to trait rumination in previous studies, but these have only used static connectivity and have yielded mixed results. The significant innovations of the current study are to use a measure of dynamic functional connectivity ("dynamic conditional correlations") and to adopt a rigorous predictive approach with validation across multiple independent datasets. Rumination is a construct that is relevant in the psychopathology of several mental illnesses, such as depression, and there is a need of generalizable neuroimaging correlates of symptoms in psychiatry. In this context, I find this manuscript rigorous and well-executed and the model that the Authors introduce, promising. Therefore, I think that this manuscript is a valuable addition to the field, and I recommend it for publication.

I am not the Reviewer 3 who made the first round of comments and in my review, I was asked to assess if the response of the Authors to previous Reviewer 3 was satisfactory. I think that the Authors have addressed the previous Reviewer's concerns well, in the following ways:

- 1) The Authors have redone the analysis using static instead of dynamic functional connectivity to show that only the variance of dynamic functional connectivity can successfully predict rumination.
- 2) The Authors have added an additional generalizability dataset comprising of two resting state runs, one of which was collected after watching a short emotional movie. The Authors show that their model successfully predicts rumination only in the resting state following the emotional

movie, raising the interesting possibility that the prediction of trait rumination might be boosted by conducting the brain scan while the participant is in a ruminative state. This is in line with previous findings, (e.g. Chen et al., 2020) and paves the way for future studies that could induce a ruminative state to obtain even better quantitative predictors of trait rumination.

3) The Authors have added text to their limitations section, acknowledging that it is unclear whether their model predicted trait rumination specifically in depressed patients and acknowledging potential reliability issues of resting state scans lasting <10 minutes.

4) The Authors have clarified the term "depressive rumination" in the Methods.

5) The Authors argue that they can't assess if thought sampling during the resting state in Study 3 induced a more ruminative state. I think that trying to do this would be highly speculative and I agree with the Authors that this should be done as part of future studies that attempt the prediction while inducing ruminations.

6) Finally, the Authors argue that the results of their virtual lesion analysis are meaningful even if only 21 regions (~40%) showed overlapping results across datasets. I agree with them, and I think that this relatively small set of regions could be the focus of future studies.

Overall, I think that the revision addresses the concerns of previous Reviewer 3 well and recommend the manuscript for publication.

Bibliography

Chen, X., Chen, N.-X., Shen, Y.-Q., Li, H.-X., Li, L., Lu, B., Zhu, Z.-C., Fan, Z., Yan, C.-G., 2020. The subsystem mechanism of default mode network underlying rumination: A reproducible neuroimaging study. *NeuroImage* 221, 117185. <https://doi.org/10.1016/j.neuroimage.2020.117185>

Ms. No.: NCOMMS-022-05713A

We would like to thank the reviewers for their thoughtful comments and constructive suggestions and the editors of *Nature Communications* for the opportunity to submit the revised manuscript. In response to the comments, we performed additional analyses and revised the manuscript as indicated in the point-by-point responses below.

Summary of main changes:

- ✧ We corrected a typo in **Table 1**, which reported the wrong correlation value in the testing dataset.
- ✧ We added two supplementary tables (**Supplementary Tables 5-6**) for comparison of model prediction results with different numbers of features.
- ✧ We changed one of the co-author's last name as she requested.

We hope our revisions satisfactorily addressed all the issues raised. Again, we appreciate all your insightful comments and the opportunity to improve the manuscript.

(Font color legends: Reviewers' comments are in **purple**, our responses are in **black**, and the revisions are in **red**.)

Reviewer #1

"I think this revision deals with all of the suggestions I made about the initial submission. I believe the paper is stronger because of these changes and as I explained in my initial review I am very excited about the paper and in particular it's convergence with other studies along similar lines. I am looking forward to seeing this paper in press in due course."

Response: We thank you so much for your thoughtful and positive comments.

Reviewer #2

"Overall the authors have answered my questions/comments from the previous review."

Response: We are glad our revision answered your questions and comments. We hope the current revision also satisfies your additional comments.

R2-1. *"I think there is an error in Table 1. For the last column (Test2 (n=48)), the r is listed as 0.24 be it should be 0.288 per the text above."*

Response: Thank you so much for pointing out our mistake. We have fixed the error in **Table 1**. We also double-checked all the numbers to make sure there were no more errors.

Revision to the manuscript p. 8 (in red):

Table1. Training and testing results of all models.

Seeds	Training (n = 84)						Test1 (n = 61)						Test2 (n = 48)	
	Brood		Depressive		Reflective		Brood		Depressive		Reflective		Depressive	
	r	p	r	p	r	p	r	p	r	p	r	p	r	p
dmPFC	-0.114	0.856	0.342	0.001	0.062	0.287			0.240	0.037			0.288	0.025
vmPFC	0.288	0.003	-0.180	0.947	0.015	0.441	0.131	0.153						
HF (L)	-0.284	0.996	0.071	0.256	-0.058	0.694								
HF (R)	0.014	0.444	-0.118	0.854	-0.170	0.937								
LTC (L)	0.052	0.321	-0.106	0.831	0.107	0.167								
PCC (L)	0.392	0.000	0.042	0.356	-0.046	0.670	-0.109	0.795						
PHC (L)	0.195	0.037	0.096	0.188	-0.008	0.531								
Rsp (L)	-0.116	0.854	0.289	0.006	0.259	0.009								
TPJ (L)	0.023	0.433	-0.007	0.520	0.221	0.019								
LTC (R)	0.166	0.065	-0.140	0.898	-0.280	0.993								
PCC (R)	-0.069	0.742	0.133	0.111	0.158	0.078								
PHC (R)	0.007	0.483	0.123	0.131	0.450	0.000					0.131	0.158		
Rsp (R)	-0.231	0.982	-0.281	0.996	-0.179	0.947								
TPJ (R)	-0.153	0.916	-0.102	0.822	0.564	0.000					0.184	0.083		
piPL (L)	0.117	0.141	-0.135	0.888	0.032	0.393								
TempP (L)	0.314	0.001	0.009	0.468	0.294	0.003	-0.045	0.631			-0.022	0.568		
aMPFC (L)	0.101	0.176	0.235	0.015	-0.165	0.933								
piPL (R)	0.126	0.131	-0.039	0.635	0.237	0.015								
TempP (R)	-0.067	0.722	-0.121	0.857	-0.114	0.854								
aMPFC (R)	0.082	0.233	0.232	0.017	0.250	0.012								

R2-2. *“It could be helpful to put in the SI the tables for combining dataset 2+3 for training (maybe one for the original number of features and one for > 80 features). That would make it easier to compare when and how replication is achieved across different training/testing breakdowns. It might be also worth noting this as a limitation.”*

Response: We agree that the results with the combined datasets 2 and 3 would be useful. Thus, we now added the results as **Supplementary Tables 5-6**. **Supplementary Table 5** shows the results from using the number of features same as the original model (i.e., $n_{feature} = 84$), and **Supplementary Table 6** shows the result from using the maximum possible number of features (i.e., $n_{feature} = 109$). Lastly, as the reviewer suggested, we now added the relevant discussion as a limitation to the Discussion section.

Revision to the manuscript p. 9

*In addition, to further investigate the impact of the number of features, we compared the results of using the number of features same as the original model (i.e., $n_{feature} = 84$; **Supplementary Table 5**) with the results of using the maximum possible number of features (i.e., $n_{feature} = 109$; **Supplementary Table 6**). In both cases, only the dmPFC-based predictive model of depressive rumination showed significant predictions across training and testing datasets.*

Revision to the manuscript p. 20

Sixth, the model’s generalizability was affected by which datasets were used in the training

procedure (**Supplementary Tables 3 and 4**). Our additional analyses suggested that it could be due to 1) differences in the distribution of the dependent variables across datasets (**Supplementary Fig. 3**) or 2) the minimum number of input features required for the generalizable prediction of rumination (**Supplementary Fig. 4, Supplementary Tables 5-6**). However, it is difficult to know whether there is a specific number of features required for successful prediction. Also, it is possible that multiple modeling options, such as input features (e.g., connectivity vs. activity), resolution (e.g., voxel-level vs. region-level), etc., interact with the required number of features. Future studies should examine these influences in more detail.

Supplementary Table 5. Training and testing results using the number of features same as the original model, $n_{feature} = 84$

Seeds	Training ($n = 109$; Studies 2 and 3)						Test ($n = 84$; Study 1)			
	Brood		Depressive		Reflective		Brood		Depressive	
	corr	permP	corr	permP	corr	permP	corr	permP	corr	permP
dMPFC	-0.016	0.561	0.275	0.002	-0.069	0.767			0.190	0.042
vMPFC	0.255	0.003	-0.154	0.948	0.186	0.029	-0.150	0.916		
HF (L)	-0.045	0.687	-0.117	0.885	0.001	0.487				
HF (R)	-0.169	0.961	0.024	0.411	-0.009	0.539				
LTC (L)	0.018	0.420	0.388	0.000	-0.117	0.891			-0.171	0.940
PCC (L)	0.034	0.366	0.032	0.367	0.206	0.017				
PHC (L)	-0.001	0.509	0.172	0.035	0.006	0.470				
Rsp (L)	0.264	0.002	-0.040	0.654	0.106	0.138	0.203	0.034		
TPJ (L)	-0.014	0.557	-0.062	0.741	-0.112	0.875				
LTC (R)	-0.014	0.551	-0.068	0.756	-0.150	0.942				
PCC (R)	0.089	0.177	-0.020	0.570	-0.025	0.604				
PHC (R)	0.104	0.142	-0.046	0.683	-0.192	0.978				
Rsp (R)	-0.217	0.988	-0.014	0.549	0.063	0.258				
TPJ (R)	-0.001	0.505	0.125	0.101	0.015	0.437				
pIPL (L)	0.101	0.152	-0.091	0.820	-0.042	0.669				
TempP (L)	-0.086	0.808	0.145	0.072	0.091	0.178				
aMPFC (L)	0.094	0.161	0.089	0.181	0.001	0.500				
pIPL (R)	0.054	0.291	-0.006	0.522	-0.038	0.650				
TempP (R)	0.068	0.241	0.299	0.000	-0.014	0.559			0.143	0.095
aMPFC (R)	0.131	0.082	0.039	0.341	0.049	0.301				

Note. This table shows the results using the combined dataset of Studies 2 and 3 ($n = 109$) as the training dataset and the Study 1 dataset ($n = 84$) as the testing dataset. We used the same 20 default mode network regions-of-interest as in Table 1 and trained and tested the models using the variance of seed-based dynamic functional connectivity. We corrected for multiple comparisons with the false discovery rate ($p < .003$ for FDR $q < .05$).

Supplementary Table 6. Training and testing results using the maximum number of features, $n_{feature} = 109$

Seeds	Training ($n = 109$; Studies 2 and 3)						Test ($n = 84$; Study 1)					
	Brood		Depressive		Reflective		Brood		Depressive		Reflective	
	corr	permP	corr	permP	corr	permP	corr	permP	corr	permP	corr	permP
dMPFC	0.066	0.255	0.244	0.005	-0.089	0.821			0.207	0.029		
vMPFC	0.189	0.026	-0.155	0.949	0.135	0.087						
HF (L)	-0.084	0.814	-0.118	0.888	0.016	0.431						
HF (R)	-0.136	0.923	-0.012	0.551	0.000	0.506						
LTC (L)	0.068	0.237	0.391	0.000	-0.029	0.625			-0.175	0.944		
PCC (L)	0.077	0.211	-0.016	0.563	0.255	0.004					0.161	0.077
PHC (L)	0.011	0.460	0.176	0.030	0.069	0.235						
Rsp (L)	0.273	0.001	-0.030	0.625	0.107	0.130	0.159	0.075				
TPJ (L)	-0.021	0.587	-0.040	0.661	-0.096	0.834						
LTC (R)	-0.044	0.668	-0.092	0.829	-0.109	0.871						
PCC (R)	0.102	0.143	-0.080	0.792	-0.037	0.650						
PHC (R)	0.073	0.226	-0.124	0.904	-0.166	0.962						
Rsp (R)	-0.231	0.992	-0.052	0.700	0.046	0.317						
TPJ (R)	0.008	0.464	0.182	0.029	-0.093	0.831						
pIPL (L)	0.065	0.256	-0.020	0.577	-0.105	0.864						
TempP (L)	-0.106	0.856	0.119	0.113	0.022	0.416						
aMPFC (L)	0.098	0.152	0.040	0.336	-0.019	0.579						
pIPL (R)	0.076	0.212	-0.049	0.694	-0.023	0.595						
TempP (R)	0.121	0.105	0.327	0.000	-0.009	0.534			0.066	0.271		
aMPFC (R)	0.160	0.046	0.087	0.183	0.083	0.197						

Note. Same as Supplementary Table 5, except this shows the results of using the maximum possible number of features for prediction.

R2-3: *“I really appreciate the tables. They make it easy to compare across models.”*

Response: We are glad that our tables are helpful for you to understanding our findings. We now added additional tables requested in **R2-2**, and hope those tables are helpful.

Reviewer #4

R4-1: *“In their manuscript, Kim et al. present a multivariate predictive model of trait rumination using as input dynamic functional connectivity between the dorsomedial prefrontal cortex and the rest of the brain. Notably, the performance of this model was tested in multiple independent datasets and the results generalized in many of them.*

Resting state fluctuations in regions belonging to the default mode network, such as the dorsomedial prefrontal cortex, have been shown to be related to trait rumination in previous studies, but these have only used static connectivity and have yielded mixed results. The significant innovations of the current study are to use a measure of dynamic functional connectivity (“dynamic conditional correlations”) and to adopt a rigorous predictive approach with validation across multiple independent datasets. Rumination is a construct that is relevant in the psychopathology of several mental illnesses, such as

depression, and there is a need of generalizable neuroimaging correlates of symptoms in psychiatry. In this context, I find this manuscript rigorous and well-executed and the model that the Authors introduce, promising. Therefore, I think that this manuscript is a valuable addition to the field, and I recommend it for publication.”

I am not the Reviewer 3 who made the first round of comments and in my review, I was asked to assess if the response of the Authors to previous Reviewer 3 was satisfactory. I think that the Authors have addressed the previous Reviewer’s concerns well, in the following ways:

- 1) The Authors have redone the analysis using static instead of dynamic functional connectivity to show that only the variance of dynamic functional connectivity can successfully predict rumination.*
- 2) The Authors have added an additional generalizability dataset comprising of two resting state runs, one of which was collected after watching a short emotional movie. The Authors show that their model successfully predicts rumination only in the resting state following the emotional movie, raising the interesting possibility that the prediction of trait rumination might be boosted by conducting the brain scan while the participant is in a ruminative state. This is in line with previous findings, (e.g. Chen et al., 2020) and paves the way for future studies that could induce a ruminative state to obtain even better quantitative predictors of trait rumination.*
- 3) The Authors have added text to their limitations section, acknowledging that it is unclear whether their model predicted trait rumination specifically in depressed patients and acknowledging potential reliability issues of resting state scans lasting <10 minutes.*
- 4) The Authors have clarified the term “depressive rumination” in the Methods.*
- 5) The Authors argue that they can’t assess if thought sampling during the resting state in Study 3 induced a more ruminative state. I think that trying to do this would be highly speculative and I agree with the Authors that this should be done as part of future studies that attempt the prediction while inducing ruminations.*
- 6) Finally, the Authors argue that the results of their virtual lesion analysis are meaningful even if only 21 regions (~40%) showed overlapping results across datasets. I agree with them, and I think that this relatively small set of regions could be the focus of future studies.”*

Overall, I think that the revision addresses the concerns of previous Reviewer 3 well and recommend the manuscript for publication.”

Response: We sincerely appreciate your careful reading of our paper and positive comments on our revision.